# Integration of 3D-printed cerebral cortical tissue into an ex vivo lesioned brain slice

Yongcheng Jin[1], Ellina Mikhailova[1], Ming Lei ®[2], Sally A. Cowley ®[3], Tianyi Sun[2], Xingyun Yang[1], Yujia Zhang ®[1], Kaili Liu[4], Daniel Catarino da Silva[4], Luana Campos Soares[4], Sara Bandiera[4], Francis G. Szele ®[4] ✉, Zoltán Molnár ®[4] ✉, Linna Zhou ®[1,5] ✉ & Hagan Bayley ®[1] ✉

Engineering human tissue with diverse cell types and architectures remains challenging. The cerebral cortex, which has a layered cellular architecture composed of layer-specific neurons organised into vertical columns, delivers higher cognition through intricately wired neural circuits. However, current tissue engineering approaches cannot produce such structures. Here, we use a droplet printing technique to fabricate tissues comprising simplified cerebral cortical columns. Human induced pluripotent stem cells are differentiated into upper- and deep-layer neural progenitors, which are then printed to form cerebral cortical tissues with a two-layer organization. The tissues show layer-specific biomarker expression and develop a structurally integrated network of processes. Implantation of the printed cortical tissues into ex vivo mouse brain explants results in substantial structural implant-host integration across the tissue boundaries as demonstrated by the projection of processes and the migration of neurons, and leads to the appearance of correlated Ca²⁺ oscillations across the interface. The presented approach might be used for the evaluation of drugs and nutrients that promote tissue integration. Importantly, our methodology offers a technical reservoir for future personalized implantation treatments that use 3D tissues derived from a patient's own induced pluripotent stem cells.

Tissue regenerative therapies have gained tremendous recent interest and promise to provide alternative treatments for a wide range of difficult-to-treat injuries and diseases. The emergence of human induced pluripotent stem cells (hiPSCs) has the potential to generate the cell types that make up all human tissues[1]. Importantly, autologous transplantation of iPSC-derived cells can minimise the immune response[2]. Here, we focus on the generation of neural tissues with laminar structures for potential applications involving implantation. Brain injuries, which include traumatic brain injury (TBI)[3], stroke[4], surgical resection for cancer[5], and epilepsy[6], can result in significant damage to the cerebral cortex, leading to a range of symptoms, including cognitive dysfunction[7-10], motor impairment[11-14] and difficulty communicating[15-18], and burdens for society. For example, it was reported in 2018 that 69 million people globally suffer from TBI and 4.8 million of these cases are severe or fatal[3,19-21]. However, effective therapeutics are still absent for the treatment of brain injuries[22]. The implantation of neural progenitor cells and brain organoids into mice has been attempted for the repair of brain injuries[23,24]. However,

[1]Department of Chemistry, University of Oxford, Oxford OX1 3TA, UK. [2]Department of Pharmacology, University of Oxford, Oxford OX1 3QT, UK. [3]James and Lillian Martin Centre for Stem Cell Research, Sir William Dunn School of Pathology, University of Oxford, South Parks Road, Oxford OX1 3RE, UK. [4]Department of Physiology, Anatomy and Genetics, University of Oxford, Oxford OX1 3PT, UK. [5]Ludwig Institute for Cancer Research, Nuffield Department of Medicine, University of Oxford, Oxford OX3 7DQ, UK. ✉e-mail: francis.szele@dpag.ox.ac.uk; zoltan.molnar@dpag.ox.ac.uk; linna.zhou@chem.ox.ac.uk; hagan.bayley@chem.ox.ac.uk

the structure of the damaged brain tissue was not fully restored in these studies because the implanted dissociated cells or organoids did not provide the cellular architecture resembling natural brain anatomy. The cerebral cortex typically has a six-layer architecture composed of layer-specific neurons. Layers I-IV are designated upper layer, while Layers V–VI are the deep layers. Intracortical wiring of neural circuits between different layers[25,26], is believed to play an important role in higher cognition in mammals[27,28]. Rather than the implantation of dissociated hiPSC-derived cells or organoids, which lack structural control, we suppose that the implantation of tissues resembling the cellular architecture of the damaged tissue will offer more effective treatments. Here, we report a droplet printing technique that produces two-layered simplified model of cerebral cortical columns. These constructs were implanted into lesions in ex vivo mouse brain explants. The implants underwent structural integration and showed correlated $Ca^{2+}$ oscillations, demonstrating a significant advance in tissue engineering en route to organ repair.

In brief, we first differentiated hiPSCs into two subtypes of neural progenitors (NPs), upper- and deep-layer neural progenitors (UNPs and DNPs; Fig. 1a, left column). These layer-specific NPs were then printed into layered cerebral cortical tissues using our 3D droplet printing technique, which enables the production of structurally defined and scaffold-free soft tissues composed of cells and extracellular matrix (ECM; Fig. 1a, middle column)[29–31]. The printed progenitor cells underwent maturation, including terminal differentiation, process outgrowth and migration. The layered structure was maintained in vitro and naturalistic layer-specific markers were expressed. The printed tissues were then implanted into lesions within ex vivo mouse brain explants (Fig. 1a), and cellular morphology, structural integration and calcium ion activity were monitored over a week.

## Results

### Layered structures by 3D-droplet printing

The 3D droplet printer contains a piezo driver which generates mechanical pulses and ejects aqueous droplets from a printing nozzle into oil (Supplementary Fig. 1a)[30]. The ejected droplets, containing ECM only (Fig. 1b) or ECM and cells (Fig. 1c, d), spontaneously acquire a lipid monolayer at the droplet/ oil interface, and contacting droplets form droplet-interface bilayers (DIB; Fig. 1a, middle, and Fig. 1b)[31,32]. With computer-aided printing, the cell-containing droplets (diameter ~100 µm) can be patterned to produce various designs of droplet networks. For example, a $8 \times 8 \times 8$ droplet network with RFP-labelled cells (Fig. 1e, f and Supplementary Fig. 1c, d) and a $12 \times 12 \times 12$ droplet network with GFP-labelled cells enveloped by RFP-labelled cells (Fig. 1g) were printed. To produce material for implantation, we printed layer-specific neurons, RFP-labelled UNPs and non-labelled DNPs (see below), into two $8 \times 8 \times 8$ droplet networks, which were brought together in a two-layer structure (Fig. 1a bottom, h–j) with a height of ~1000 µm and width of ~500 µm, to form a simplified version of a cerebral cortex column comprising upper-layer and deep-layer segments[33]. Raising the temperature, from room to physiological, facilitated gelation of the ECM and annealing of printed two-layer networks, containing either cells (Supplementary Fig. 1d) or microbeads (Supplementary Fig. 1e). A flow chart of the printing process is shown in Supplementary Fig. 2.

To demonstrate that our printing technique might produce structures representing all six layers of the cerebral cortex[33], we applied a layer-by-layer sequential printing strategy (Supplementary Fig. 1f). Each layer, composed of an $8 \times 8 \times 8$ droplet network, was labelled with a different colour (Fig. 1k and Supplementary Fig. 1g). In addition to printing sub-millimetre scale cubic structures, centimetre-scale structures with diverse shapes were also printed (Fig. 1l–n and Supplementary Fig. 1h).

## Generation of layer-specific neural cells

The generation of layer-specific cerebral cortical progenitor cells from hiPSCs was the essential first step towards fabrication of the two-layer cortical tissue. In humans and most other mammals, the layers of the cortex are formed in an inside-first-outside-last order. Deep-layer neurons, as the early product of cortical neurogenesis, divide asymmetrically from radial glia cells (RGCs) and migrate toward the cortical plate by radial migration from the ventricular zone[34]. Similarly, recent differentiation protocols have reported that, in early cultures, NPs primarily differentiate into deep-layer neurons (DNs), which can be phenotyped by the expression of the deep-layer marker CTIP2[35,36].

To generate DNs, we applied a dual-SMAD inhibition differentiation method[37] to generate NPs in monolayer culture. Human iPSCs, reprogrammed from a healthy individual's somatic cells[37], were confirmed for their pluripotency. Immunostaining showed high expression of the pluripotency markers TRA-1-60, NANOG and OCT4 (Fig. 2b, left column and Supplementary Fig. 3a). We used a defined neural induction medium (NIM), containing the two SMAD inhibitors, LDN193189 and SB431542, to induce the hiPSCs into neural ectoderm lineage. The neural ectoderm cells were then cultured in neural maintenance medium (NMM), which enables the generation of NPs by 19 Days in vitro (DIV19). Further maturation was achieved by seeding NPs at a low density (100,000 cells/cm²) and using neural terminal medium (NTM) containing the γ-secretase inhibitor (DAPT), which blocks the presenilin–γ-secretase complex[38] and prevents the downstream activation of Notch[39]. The inhibition of the Notch pathway switches the differentiation from glial to neuronal cell fates[40]. DAPT has been applied in various hiPSC differentiation protocols to facilitate neuron differentiation and maturation[41–43]. After 10 days of culture in NTM, DNs showed the features of mature neurons at DIV 29+ with a polarised morphology and the expression of the deep-layer marker CTIP2, but low expression of middle-upper (SATB2) and upper layer (CUX1 and BRN2) markers (Supplementary Fig. 4a, b).

Later in cortical neurogenesis, RGCs generate neurons that migrate radially into the cortical plate, passing through the DNs, to become upper-layer neurons (UNs)[34,44]. A recent protocol from Boissart et al. addressed the generation of UNs in vitro through prolonged pro-proliferative culture of NPs, which resembles the in vivo process of the late production of UNs from proliferating RGCs[43]. Following this protocol, we conducted an extended treatment of NPs with a growth factor cocktail, resembling the key steps of Boissart's protocol[43]. In contrast with the original protocol, we adopted small molecules for induction and the monolayer system in our neural differentiation protocol to reduce batch-to-batch variation. The growth factor cocktail included a combination of growth factors that support proliferation (FGF-2 and EGF)[45], and survival and maturation (BDNF)[43,46]. During the treatment, the cells retained progenitor morphology, and underwent 8–10 doublings from the beginning of neural induction until DIV 40. A representative culture on DIV 31 is shown in Fig. 2b (middle column). The UNPs at DIV 40 could then be harvested for further maturation culture or cryopreservation. To continue maturation of UNPs, the growth factors were withdrawn at DIV 40, and a further incubation in NTM containing DAPT over ten days was conducted to generate mature UNs at DIV 50+ (Fig. 2b, right column and Supplementary Fig. 5a–c). Without DAPT, however, the cells showed a non-polarised morphology which was similar to the UNPs, indicating failure to undergo maturation (Supplementary Fig. 5a). The UNs were morphologically similar to DNs (Supplementary Fig. 5a and Fig. 4a). However, immunofluorescent staining of DIV50+ UNs showed the expression of upper-layer markers CUX1, CUX2 and BRN2, and the middle-upper layer marker SATB2, whereas CTIP2 expression was rarely detected (Fig. 2c and Supplementary Fig. 5d). A quantitative analysis showed that DIV50+ UNs expressed CUX1, BRN2 and SATB2 at 68 ± 8%, 74 ± 7%

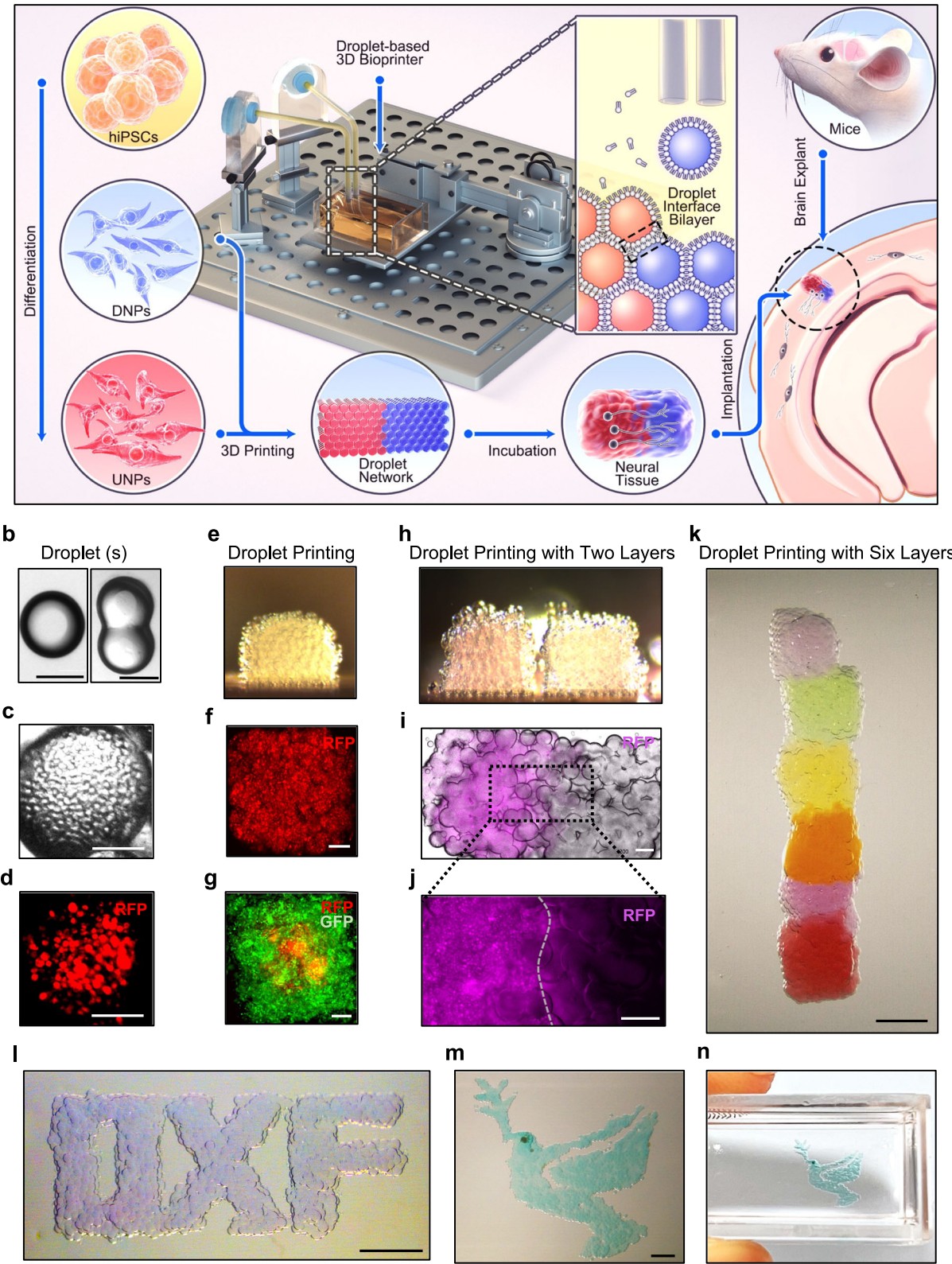

**a**

**b** Droplet (s)

**c**

**d** RFP

**e** Droplet Printing

**f** RFP

**g** RFP GFP

**h** Droplet Printing with Two Layers

**i** RFP

**j** RFP

**k** Droplet Printing with Six Layers

**l**

**m**

**n**

and 70 ± 7%, respectively. In contrast, only 16 ± 4% cells expressed deep-layer marker CTIP2 (Fig. 2d).

To further confirm the identities of the DN and UN cells, we conducted gene expression analysis using a Real-time Quantitative Polymerase Chain Reaction (RT-qPCR). The treatment with the growth factor cocktail significantly upregulated CUX1 expression with a 20- and 22-fold change in DIV40+ UNPs and DIV50+ UNs compared to hiPSCs, whereas no upregulation was detected in DIV19 DNPs. In addition, we did not detect CTIP2 expression changes in DIV40 UNPs. By comparison, DIV 47+ UNs in Boissart's original protocol also

**Fig. 1 | Droplet-based 3D bioprinting. a** Overview of the study. Patterned 3D printing of droplets containing the hiPSC-derived neural progenitors, deep-layer neural progenitors (DNPs) and upper-layer neural progenitors (UNPs), and extracellular matrix (ECM). The formation of adhesive DIBs secured the patterned network. The printed cerebral cortical tissues were cultured in vitro for functional studies and implanted into mouse brain explants. **b** Bright-field images of a single droplet in oil (left) and a pair of droplets connected through a DIB (right). The droplets contain solidified ECM. **c, d** Image of a droplet in oil containing RFP-labelled DNPs in ECM. **e** Side-view of an 8x8x8 printed droplet network containing DNPs. **f** Image of a printed droplet network containing RFP-labelled DNPs. **g** Image of a patterned droplet network containing GFP-labelled 3T3 cells (outer compartment) and RFP-labelled MDA breast cancer cells (centre compartment). **h** Side-view of two 8x8x8 printed droplet networks containing two layers (left and right). **i** Image of a printed two-layer droplet network containing RFP-labelled UNPs (left) and unlabelled DNPs (right). **j** Fluorescence image of a section of 'i' (indicated by the dashed box) at higher magnification. **k** Image of a printed 6-layered droplet network resembling the structure of a cortical column. **l–n** Images of centimetre-sized droplet networks. Scale bars: **b–d**, 100 μm; **f, g, i** and **j**, 200 μm; **k–n**, 1000 μm.

revealed ~20-fold upregulation of CUX1, along with a ~10-fold increase in CTIP2 expression[43]. We identified PAX6, a neural stem cell marker[47], in DIV19 DNPs indicating successful cortical neural induction. The expression of PAX6 was decreased in DIV40 UNPs and dropped further after maturation in DIV50+ UNs, with ~160-, 12- and 3-fold expression compared with hiPSCs in DIV19 DNPs, DIV40 UNPs and DIV50+ UNs, respectively. NESTIN, a neural marker, was upregulated over the differentiation and maturation process from a low level in DIV19 DNPs, and showed a 2-fold increase in DIV40 UNPs and a 7-fold increase in DIV50+ UNs compared with hiPSCs (Fig. 2e)[48].

By adapting the reported protocols[36,43], we produced two distinct progenitors, DNPs and UNPs, which gave rise to the corresponding mature layer-specific neurons: DNs and UNs. Our DIV 29+ DNs expressed the deep-layer marker CTIP2 but rarely expressed upper-layer markers. This result is consistent with reported protocols. For example, Shi's protocol producing ~40% CTIP2- and 5% BRN2-expressing neurons at DIV30[36]. By comparison, Boissart's protocol generated over 70% of UNs expressing upper-layer markers CUX1 and BNR2 on DIV 47+[43], consistent with the DIV 50+ UNs in this study (68% expressed CUX1 and 74% expressed BRN2).

**Printing two-layer cerebral cortical tissue**

To fabricate two-layer cortical tissues, the progenitor cells, DNPs and UNPs, were harvested for printing. Progenitors, instead of mature neurons, were used because the progenitors were less sensitive to the dissociation procedure from 2D cultures compared to mature neurons and were compact for printing. We printed the tissues in oil, followed by phase transfer into growth medium. For the first week of post-printing culture (WPP), the cortical tissues were incubated in NMM supplemented with a growth factor cocktail (FGF-2, EGF and BDNF) to facilitate tissue survival. At the end of 1 WPP, the growth-factor supplemented NMM was replaced with NTM to encourage the maturation of the cortical tissues. The tissues were harvested at 2, 4 and 8 WPP and assessed for morphology, cell migration, process outgrowth and gene expression (Fig. 3a).

We first fabricated deep-layer cortical tissues (8 × 8 × 8 droplet networks) from DNPs (Fig. 3b, top). The printed tissues were incubated in NMM supplemented with growth factors (Fig. 3b, middle and Supplementary Fig. 6a, left). A representative deep-layer cortical tissue at 8 WPP was sectioned and immunostained to reveal the tissue structure and the cellular composition, which was visualised with neural markers: stem cell marker SOX2, general marker of young neurons TUJ1, deep-layer markers (CTIP2 and TBR1), and upper-layer marker (CUX1; Fig. 3b bottom, c and Supplementary Fig. 6b). CTIP2/TBR1-expressing DNs and sparse CUX1-expressing UNs were observed, which are comparable with in vitro differentiated human cortical neurons[36] and brain organoids[49]. Upper-layer cortical tissue at 8WPP, in contrast, demonstrated CUX1-expressing UNs and sparse CTIP2/TBR1-expressing DNs (Supplementary Fig. 6b).

To generate cortical tissues with two layers, we printed two 8 × 8 × 8 droplet networks side-by-side, one containing DNPs and the other UNPs (Fig. 3d, top). We expected that the DNPs and UNPs would further differentiate during post printing culture to give the corresponding mature neurons, DNs and UNs, in the layers where they were printed. To determine (1) whether the two layers were preserved during post printing culture; and (2) whether the DNPs and UNPs were converted to DNs and UNs, we characterised the printed tissues at different differentiation time points.

Neuronal process outgrowth and migration are two important developmental phenomena of cortical neurogenesis. After 2-weeks of culture, the printed cortical tissues remained in the desired two-layered architecture, as illustrated by bright-field images (Supplementary Fig. 6a, right) and images with fluorescence from nucleus staining (DAPI, all cells) and RFP-labelled UNs (Fig. 3d, middle). Further immunostaining of sectioned 2 WPP cortical tissues revealed that the majority of cells in both layers expressed the neuronal marker TUJ1 and the neural stem cell marker SOX2 (Fig. 3d, bottom), indicating that the cells in both layers were neural. Z-projection images of the two-layered tissues at 8 WPP revealed that most of the neurons had acquired a polarised morphology with long processes, suggesting that neural differentiation and maturation had occurred in the printed tissue (Fig. 3e). A magnified view showed that the upper-layer neurons had produced processes projecting toward the deep layer (Fig. 3e). Neuron migration between layers was also found in the printed tissues, indicated by arrows in Fig. 3e and Supplementary Fig. 6c. A movie of the 3D-reconstructed cortical tissue further illustrated the abundance of processes that crossed between layers (Supplementary Movie 1). Comparison of the two-layer tissues at 2, 4 and 8 WPP revealed the dynamics of cross-layer process outgrowth and neuron migration (Fig. 3f). Quantitative analysis showed significant migration of RFP-labelled UNs into the deep-layer over 8 weeks of incubation, but no significant change of RFP-signal was found in the upper layer (Fig. 3g and Supplementary Fig. 6d).

Next, we assessed the spatial-temporal expression of general and layer-specific neural biomarkers to reveal the dynamics of neural maturation during post printing culture. Immunostaining of sectioned tissues at the three time points (2, 4 and 8 WPP) revealed a significantly higher population of cells in the upper layer compared to the deep layer that expressed the upper-layer marker CUX1 (78 ± 2% vs 37 ± 3 %, mean value over 2, 4 and 8 WPP) and the middle-upper layer marker SATB2 (68 ± 4% vs 39 ± 4%). Conversely, a higher percentage of cells in the deep layer compared to the upper layer expressed the deep-layer marker CTIP2 (35 ± 4% vs 10 ± 2%) at 2 WPP. The percentages of cells expressing CTIP2 in the deep and upper layer dropped to 18 ± 5% and 5 ± 0.2% respectively at 4 WPP and further fell at 8 WPP (4 ± 2% and 3 ± 0.6%, respectively). This result is consistent with the previously reported observation in cerebral organoids where the population of cells expressing CTIP2 decreased from 30 to 105 days in culture after an initial increase during the first month, mimicking the temporal patterning observed in the mouse[50]. In addition, we found that a substantial population of cells (>90%) expressed neuronal marker TUJ1 at all time points in both layers, indicating that the majority of cells in the printed tissues had committed to the neuronal lineage (Fig. 3h, i and Supplementary Fig. 6e, f). Together, these data demonstrate that the printed two-layer tissues retained the designed cellular architecture, with dynamic process outgrowth and cell migration. Further, the expected layer-specific marker expression was observed during the eight-week maturation process.

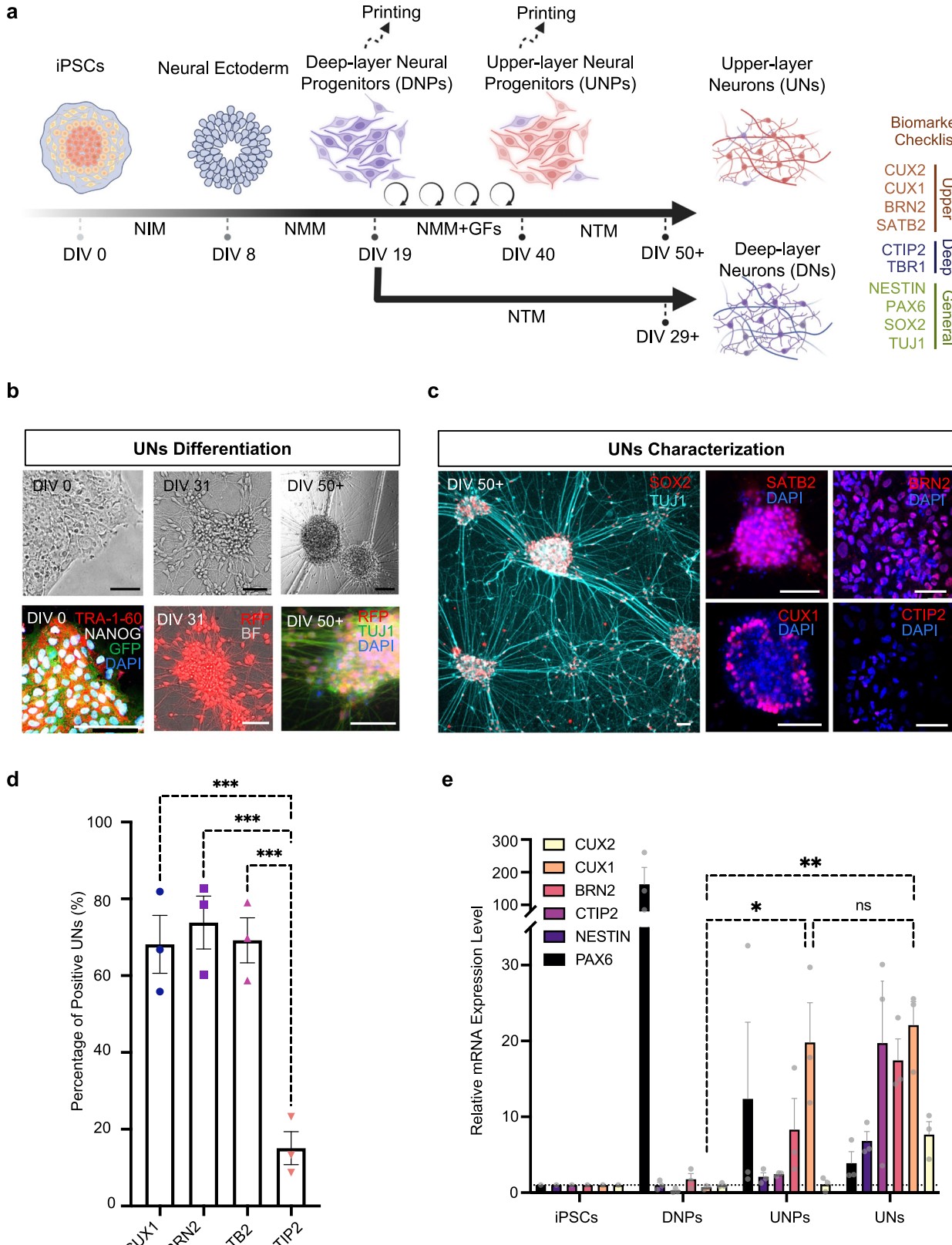

## Structural integration of printed cortical tissue with ex vivo brain explants

Cultured organotypic brain explants preserve brain architecture and cellular function in an ex vivo environment[51,52]. We implanted printed cortical tissues into lesions in the cortex of mouse ex vivo brain explants to assess their ability for tissue repair. We first printed the cortical tissues (on day -1) and cultured them for 1 day before implantation (Fig. 4a). On day 0, we prepared the brain explants and created an ~800-µm diameter circular lesion in the cerebral cortex. The explants were then cultured on Transwell inserts and the printed tissues were implanted into the lesion (Fig. 4a and Supplementary Fig. 7a, b). The implanted explants were then cultured under either

**Fig. 2 | Generation of layer-specific neural cells. a** Schematic showing the differentiation timeline of hiPSCs to DNs and UNs, via the corresponding progenitors: DNPs and UNPs. Abbreviations: Day in vitro (DIV); Neural Induction Medium (NIM); Neural Maintenance Medium (NMM), Neural Terminal Medium (NTM); Growth Factors (GFs). DIV 0–8: hiPSCs induction, committing to a neural ectoderm lineage; DIV 8–19: neural ectoderm cells differentiating into DNPs; DIV 19–40: DNPs differentiating to UNPs during extended culture in NMM supplemented with growth factors (FGF-2, EGF and BDNF). DIV 19–29+ and DIV 40–50 +, DNPs and UNPs maturing as DNs and UNs, respectively. **b** Bright-field images (top row) and immunocytochemistry images (bottom row) of cells at different stages of differentiation: DIV 0, 31, 50+, correspond to GFP-labelled hiPSCs, RFP-labelled NPs and RFP-labelled UNs respectively. **c** Immunocytochemistry analysis of the young neuron marker β3-tubulin (labelled with the TUJ1 antibody), the neural stem cell marker (SOX2), middle-upper layer marker (SATB2), upper layer markers (CUX1 and BRN2) and the deep layer marker (CTIP2) expression in differentiated RFP-labelled UNs at DIV50+. See Supplementary Fig. 4b for further immunocytochemistry analysis of DNs. **d** Quantification of marker expression in **c** ($n = 3$ biological replicates; mean ± SEM; one-way ANOVA test). **e** Quantitative RT-PCR analysis of upper layer markers (CUX1, CUX2 and BRN2), deep layer marker (CTIP2), neurofilament marker (NESTIN) and neuroectoderm marker (PAX6). Marker expression of indicated cell types relative to RFP-labelled hiPSCs. ($n = 3$ biological replicates; mean ± SEM; one-way ANOVA test). For **b** and **c**, scale bar, 50 μm. For both **d** and **e**, ns not significant. *$P < 0.05$, **$P < 0.01$ and ***$P < 0.001$.

condition A or B for one day, followed by DAPT treatment for 4 days (see Supplementary Table 1). Condition A uses a nutrient-enriched formula modified from DMEM/F12 and Neurobasal medium containing a high glucose level of ~25 mM[53], whereas condition B primarily uses a commercially available BrainPhys medium with a physiologically relevant glucose level of ~2.5 mM. The high glucose concentration can inhibit neuron differentiation through oxidative and endoplasmic reticulum stress[54]. Previous reports have indicated the superior performance of BrainPhys medium compared to DMEM/F12 and Neurobasal medium on the 2D culture of hiPSC-derived neurons by supporting neuronal survival and function, as manifested by frequent action potential firing and long-term electrical activity[53]. To evaluate the effects of condition A, condition B and DAPT on the integration of the implant, process outgrowth and neuron migration from the implants into the host tissue were measured (Fig. 4b).

Fluorescence confocal imaging revealed process outgrowth and neuron migration from the implant towards the host, indicating that the printed tissues had integrated into the brain explant (Condition B, no DAPT; Fig. 4c). Live/dead staining showed that the cells in the brain explants were 86 ± 3% viable at 5 days post-implantation (DPIs; Supplementary Fig. 7c, d). The viability of the implants was similar as indicated by their RFP expression (Fig. 4c). Analysis of the RFP-labelled neurons also revealed process outgrowth and neuron migration from 1 to 5 DPIs (Fig. 4d). Profile plots of fluorescence intensity revealed the extent of process outgrowth at 1DPI (220 μm) and 5 DPI (405 μm; Fig. 4e).

Process outgrowth and neuron migration is regulated by microenvironmental cues during neurogenesis[55,56]. We, therefore, hypothesised that process outgrowth and cell migration might respond to different nutrient conditions and small molecular treatment, such as DAPT. To address this, we cultured implanted explants under four conditions: condition A and B, with and without DAPT. Using fluorescent confocal imaging, we found differences in the distance of outgrowth and migration under the four conditions at 5DPIs (Fig. 4f). Quantitative analysis showed a significant increase in the distance of process outgrowth in condition B (434 ± 41 μm) compared to condition A (265 ± 30 μm). Interestingly, DAPT treatment led to further increases in the distance for both condition A (increased by 153 μm to 418 ± 70 μm) and condition B (increased by 287 μm to 721 ± 71 μm; Fig. 4g and Supplementary Fig. 7e). Therefore, our data illustrated that the implanted explants can be used to evaluate the effect of nutrients and small molecule treatment on implantation.

We also examined the influence of the duration of the pre-implantation incubation of the printed tissues. We compared the distance of process outgrowth and neuron migration from the implants with 1- or 14-days pre-implantation incubation in condition B. The outcome suggested that a 14-day pre-implantation culture of cortical tissues can extend implant-to-host process outgrowth (Fig. 4h). We found an increase in process outgrowth of 222 μm to 656 ± 63 μm at 5 DPIs, compared with the implants that had undergone a 1-day pre-implantation incubation (Fig. 4i and Supplementary Fig. 7e).

Under high magnification, we identified individual neurons migrating across the implant-host boundary (Fig. 4j). Quantitative analysis was conducted by counting the number of RFP-labelled neurons in an area between 200 and 400 μm away from the implant. The analysis showed 20 ± 2, 17 ± 3 and 18 ± 3 RFP-labelled neurons/0.1 mm² migrated into the host brain explant at 5 DPIs from implants composed of UNs, DNs, and 14 day pre-cultured DNs respectively. The migration of DNs (0.7 ± 0.4, 13 ± 3, 17 ± 3 neurons/0.1 mm² at 1, 3 and 5 DPI) and UNs (1.4 ± 0.4, 12 ± 4, 20 ± 2 neurons/0.1mm² at 1, 3 and 5 DPI) were observed over three time points (Fig. 4k).

## Ca²⁺ activity of printed two-layer cortical tissue in ex vivo brain explants

Here, we printed tissues with a size compatible with the cerebral cortex of P8 mice (800–1000 μm)[57] and a simplified laminar architecture consisting of deep and upper layers. The two-layered cortical tissue was then implanted into the lesioned cortex of a mouse ex vivo brain explant. Importantly, the orientation of the implanted tissues was matched with the cortex of the host; the deep layer of the implant was implanted into the ventral region of the cortex and the upper layer into the dorsal region. Integration between the implants and host was evaluated through the extent of process outgrowth and neuron migration from the implant towards the host (Fig. 5a, c and Supplementary Fig. 7f). Quantitative analysis of process outgrowth and neuron migration at 1, 3 and 5 DPIs showed an increase in process projection distance over the culture periods: 85 ± 14 μm, 336 ± 22 μm to 419 ± 22 μm, respectively (Fig. 5b). Further, immunostaining showed human-specific neural marker HNCAM expression in both layers confirming the human origin of the implanted neurons, while RFP expression was sharply lower in the upper layer compared to the deep layer, indicating that the two-layer pattern had been maintained in the implant (Fig. 5d, e).

In the nervous system, correlated Ca²⁺ oscillations of cells in neuronal circuits are necessary for brain functions[58–60]. To evaluate the activity of the implanted cortical tissues, we performed Ca²⁺ imaging with Fluo-4, a fluorescent calcium indicator. Time-lapse recordings of implanted explants revealed spontaneous Ca²⁺ oscillations of cells in both the implant and the host at 5 DPIs (Fig. 5f). Simultaneous calcium ion fluctuations in adjacent cells, suggested potential correlation between these cells (Fig. 5g). To seek correlations of calcium oscillations between implant and host, we recorded Ca²⁺ signals at the deep-layer only implant/ explant interface (Fig. 5h and Supplementary Movie 2). By applying the correlation calculation method reported by Ko et al. [61], we found that the neurons exhibited Ca²⁺ oscillations with a correlation factor of $R > 0.1$ at 5 DPIs, suggesting communication between the implant and the host (Fig. 5i and Supplementary Fig. 8a). Additional similarity matrices and correlated network analyses supported the existence of multiple neuronal communities with correlated firing patterns (Fig. 5j and Supplementary Fig. 8b,e). Further Ca²⁺ imaging and correlated network assessment on implanted 5 DPI upper-layer only tissue (Fig. 5k, l and Supplementary Fig. 8c–e) and implanted two-layer tissue (Supplementary Fig. 8f–h) showed correlated cell

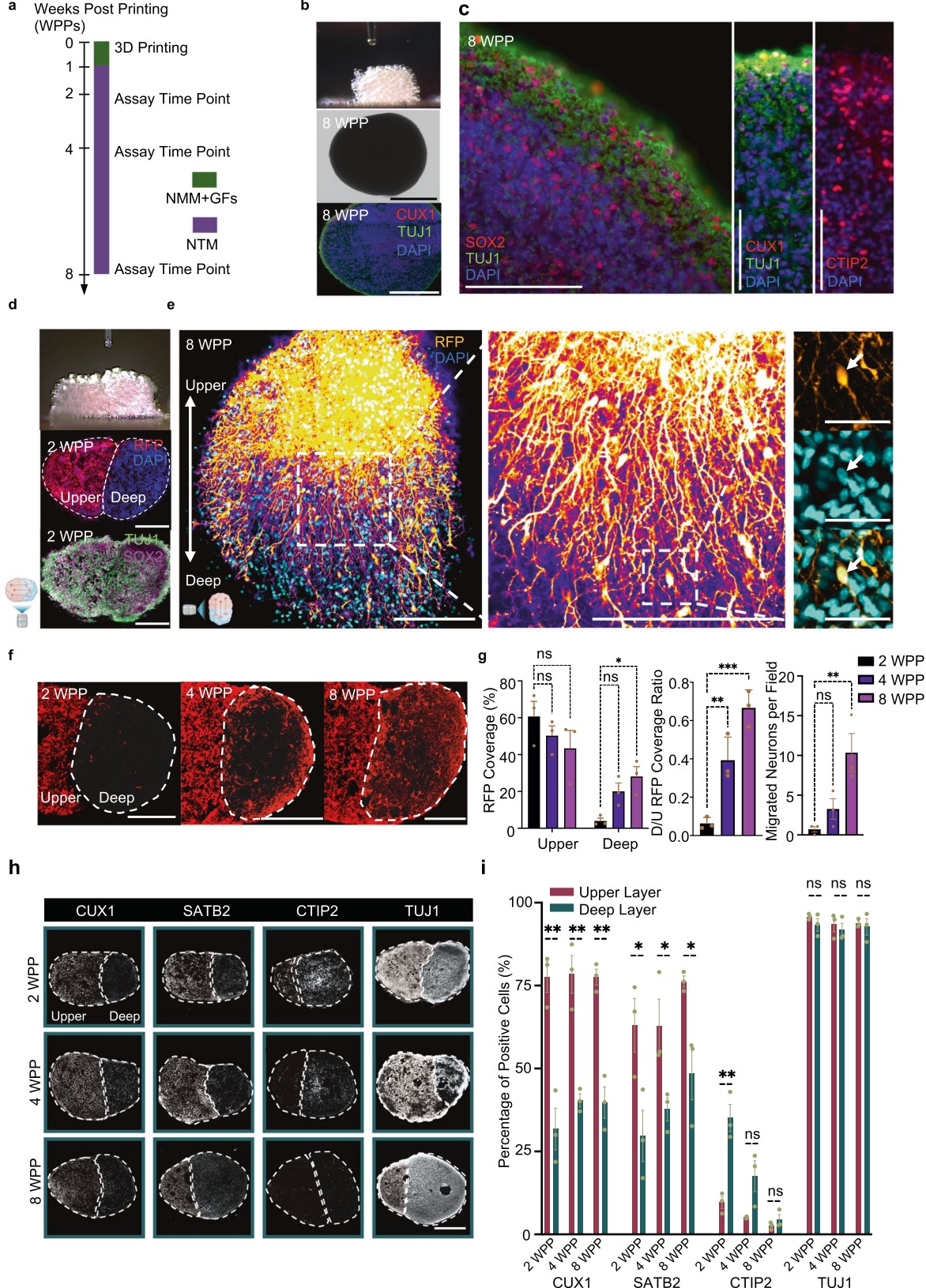

pairs within the implant and the host, and across the upper/ deep and implant/ host boundaries. Specifically, we demonstrated a group of regions of interest (ROIs) with correlated Ca²⁺ traces between implant and host (Fig. 5m). Overall, for 3–5 DPI implanted explants, we found $4.4 \pm 0.7\%$ correlated host-to-host cell pairs, $1.6 \pm 0.3\%$ host-to-implant cell pairs and $2.0 \pm 0.9\%$ implant-to-implant cell pairs (Fig. 5n).

## Discussion

Here, we demonstrate that droplet-based 3D-printing can be used to produce tissues with the architecture of a simplified two-layer cerebral cortical column. The identity and structure of the deep and upper layers were maintained during in vitro culture after printing. During this period, process outgrowth, neuron migration and maturation

**Fig. 3 | Construction of cerebral cortical tissue with deep and upper layers.**
**a** Timeline for the production and in vitro culture of droplet-printed cortical tissues. Further details are described in "Methods" section. **b** Side view of an ongoing printing process (top), bright-field image of the unlabelled deep layer cortical tissue after 8 weeks post-printing (WPP) (middle), and fluorescence immunostaining of sectioned 8 WPP tissue (bottom). **c** Fluorescence images of sectioned 8 WPP unlabelled deep-layer cortical tissue showing the expression of stem cell (SOX2), young neuronal (TUJ1) and layer-specific markers (CUX1, CTIP2 & TBR1). Scale bars, 100 μm. **d** Printing of two-layer cortical tissues. Side view of a printed two-layer cortical tissue (top), and fluorescence images of a sectioned 2 WPP two-layer tissue with RFP-labelled UNs and unlabelled DNs (middle and bottom). **e** Confocal z-projection image (left) and the magnified image (middle and right) showing cross-layer process outgrowth and neuron migration in a printed two-layer tissue at 8 WPP, visualised by RFP (false coloured as fire) expression in UNs and DAPI nucleus

staining in both UN and DNs. Arrow indicates a migrating UN. Scale bars: left and middle, 500 μm; right, 50 μm. **f** Fluorescence images of cortical tissue sections with cross-layer process outgrowth and cell migration at 2 (left), 4 (middle) and 8 (right) WPP. Dashed circles indicating the unlabelled deep-layer segment with invading RFP-labelled UNs. Scale bars, 200 μm. **g** Quantitative analysis of process outgrowth by percent RFP coverage (left and middle), and neuron migration by cell body counting (right), as shown in 'f' (n = 3 biological replicates; mean ± SEM; one-way ANOVA test). Field size, 0.1 mm². **h** Spatiotemporal immunofluorescence analysis of general neural and layer-specific marker expression on sectioned two-layer cortical tissues with RFP-labelled UNs and unlabelled DNs. Dashed line outlines the layers. **i.** Quantitative analysis of marker expression in upper and deep layers of cortical tissues in **h** (n = 3 biological replicates; mean ± SEM; two-sided unpaired Student's t test). ns = not significant; *P < 0.05; **P < 0.01; ***P < 0.001. Section thickness: 30 μm. For **b**, **d** and **h**: scale bars, 500 μm.

were observed. Implantation of the printed cortical tissues into ex vivo brain explants demonstrated the formation of structural connections and correlated Ca²⁺ oscillations between the implant and the host.

The emergence of hiPSCs and recent advances in stem cell differentiation, particularly those producing deep- and upper-layer-specific neurons[36,43] encouraged us to fabricate a layered cortical tissue in three dimensions. Although we demonstrated that our droplet-printing technique is capable of producing six-layered structures, mimicking the complex human cerebral cortex architecture, current hiPSCs techniques have not produced populations of neurons representing all six layers. Further advances in the generation of layer-specific cortical neurons, along with our droplet printing technique, will enable the fabrication of more fine-grained and realistic 3D ex vivo cortical tissues[25,26].

In the current study, we printed neural progenitors, DNPs and UNPs, instead of their mature descendants. These progenitors differentiated in the printed tissues during post printing culture. This strategy allowed us to avoid the difficulties associated with handling mature neurons which are known to be sensitive to dissociation from culture vessels and would likely be damaged in the 3D printing process due to their sensitivity to physical stress, changes in temperature, and changes in osmolarity[62].

Interestingly, after printing, the DNPs and UNPs continued to mature in the host towards DNs and UNs respectively, despite the fact that they were in a common growth medium. The use of lineage-committed progenitors represents a useful strategy for the fabrication of 3D tissues. Recently, the Lewis group reported a co-differentiating strategy in 3D tissues derived from two types of hiPSCs, transfected with either neural or endothelial-associated transcription factors (TFs), to produce vascularised neural tissues[63]. Although the study produced patterned neural tissues containing distinct cell types, it relied on lentivirus-based genetic modifications, which might have limited application potential in implantation therapies[64,65]. In our strategy, the production of neural tissues containing distinct types of neurons without genetic manipulation, reduces concerns over clinical safety, and might be applied to the construction of other tissues containing multiple cell types.

The transplantation of dissociated human neurons into the mouse brain has been reported in several studies, which have demonstrated the survival of injected cells, pathway generation and implant-host connections[23,66]. However, the transplantation of dissociated cells has not been reported to restore the architecture of lost tissue, for example, the laminar structure of the lesioned cortex. With droplet printing, an implant can be designed to emulate the dimensions, orientation, cellular composition and structure of the lost tissue. Particularly in the case of a large lesion, the implantation of a replacement tissue with matched 3D shape and cellular architecture, resembling the brain anatomy essential for higher cognition, might provide a precise treatment. In the present study, we

demonstrated a droplet-based 3D bioprinting technique, capable of generating two-layer cortical tissues that can be implanted into mouse ex vivo brain explants. In the implanted explants, we observed structural integration, based on implant-to-host process outgrowth, neuron migration, and correlated Ca²⁺ oscillations between the implant and the host. Considering the short period of post-implantation culture, the correlated Ca²⁺ oscillations may be the result of the early establishment of volume transmission, a neuronal signal transmission mechanism conducted by non-synaptic release of neurotransmitters, which diffuse through the extracellular space[67,68]. Implantation after longer post-implantation incubation times might lead to more advanced repair, an aim of our future work. Further research could take advantage of potential advances in neural differentiation that produce further layer-specific neurons and thereby more realistic cortical tissues. Progenitors derived from patients' own cells might be used to produce implants to treat currently incurable brain damage.

In the present study, our primary emphasis has been on the implantation of tissues prepared by our 3D printing technique, and only one hiPSC line, AH016-3, was tested. Nonetheless, the differentiation process was applied to these hiPSCs multiple times with similar results. The utilisation of mouse sarcoma-derived Matrigel presents potential challenges in clinical applications, such as batch-to-batch variation, limited supply, and safety concerns arising from its tumour-derived origin. Hydrogels sourced from the ECM of specific tissues have been reported as viable alternatives to Matrigel for organoid production[69]. However, the composition and structure of these hydrogels may still exhibit variation from one batch to another. To overcome the limitations associated with Matrigel or other ECM-derived hydrogels, chemically defined hydrogels for cells and organoids have been reported[70,71]. Our droplet-based method is capable of printing alternative hydrogels, such as collagen and agarose[72], suggesting potential for the future printing of 3D tissues comprising diverse and chemically defined hydrogels.

## Methods

Animal experiments were performed on mice in accordance with the United Kingdom Animals (Scientific Procedures) Act 1986 and were approved by the University of Oxford Pharmacology ethical committee (approval ref. PPL: PP8557407) in conformity with the national guidelines under which the institution operates. All mice used in this study were maintained in pathogen-free facilities at the University of Oxford. Mice were given *ad libitum* access to food and water.

Catalogue numbers of materials used in this study are in Supplementary Tables 1–3.

### hiPSC differentiation

**Cell lines.** The hiPSC lines used in this study were derived by Dr Sally Cowley (James Martin Stem Cell Facility, Oxford). We used a single

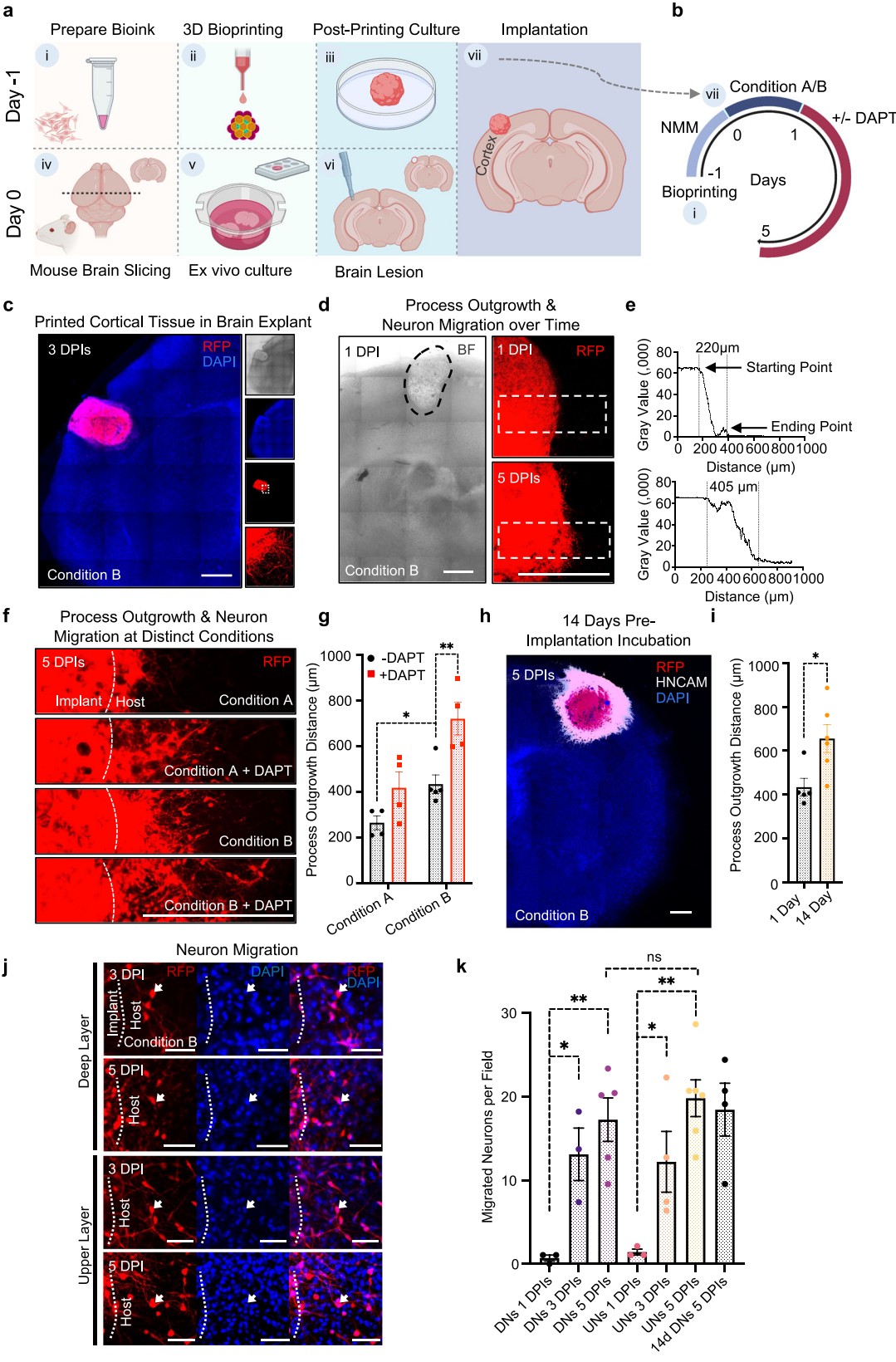

AH016-3 hiPSC line without or with fluorescent labelling: unlabelled, RFPiced-labelled and GFP-labelled[37].

**Maintenance.** iPSC lines were maintained and expanded at 37 °C in 5% CO₂ in mTeSR™Plus medium (Stemcell Technologies) on Geltrex (Thermo Fisher)-coated plates with daily medium changes and

passaging every 3-4 days with releasing agent ReLeSR (Stemcell Technologies).

**DNP differentiation.** On DIV-1, hiPSCs were passaged 1:1 or 3:2 after release as small clusters with 0.5 mM EDTA. To achieve this, cells were washed with Dulbecco's phosphate-buffered saline (DPBS;

**Fig. 4 | Implantation of printed single-layer cortical tissue into brain explants.**
**a** Steps in the implantation process. Step i, bioink prepared from DNPs or UNPs with Matrigel. Step ii, droplet-printing of cortical tissue. Step iii, post-printing culture of the printed tissue for 1 day. Step iv, preparation of mouse brain slices with a thickness of 300 μm. Step v, brain slices (explants) were kept under condition A or condition B before implantation. Step vi, creating the brain lesion with a biopsy punch. Step vii, implanting printed tissue into the lesion of the explant, which was then further incubated. **b** Incubation of the implanted explant and drug treatment. Explants were incubated under either condition A or condition B for 1 day, followed by treatment with or without DAPT for 4 days. **c** Left, tiled fluorescence confocal image of an implanted mouse brain explant at 3 days post implantation (DPI) under condition B. The implant contained RFP-labelled DNs. Right, bright field (top), DAPI staining (upper middle), RFP-labelled implanted tissue (lower middle) and higher magnification view (bottom). The culture media are described in the "Methods" section. **d** Confocal image of a representative 1 DPI implanted explant (left) and confocal fluorescence images of process outgrowth and neuron migration from an explant with RFP-labelled DNPs at 1 (top right) and 5 (bottom right) DPI. **e** Profile plots of fluorescence intensity along the white dashed boxes from left to right indicated in **d**. The vertical dashed lines indicate the margins of the red fluorescence. **f** Confocal images of implant-to-host process outgrowth and neuron migration at 5 DPI. The implanted explants with RFP-labelled DNPs were cultured under the two conditions and with or without DAPT treatment. Dashed lines indicate the original borders of the implants. **g** Quantitative analysis of process outgrowth and neuron migration distance visualised in **f**. ($n = 4,4,5,4$ biological replicates from left to right; mean ± SEM; two-sided unpaired Student's $t$ tests). **h** A tiled confocal image of an explant implanted with RFP-labelled deep-layer cortical tissue that had been cultured for 14 days prior to implantation. The implant is immunostained with the human-specific neural marker, HNCAM. RFP localisation indicates process outgrowth and neuron migration from the implanted tissue. **i** Quantitative analysis of process outgrowth and neuron migration from deep-layer cortical tissue cultured for 1 day or 14 days prior to implantation ($n = 5, 6$ biological replicates from left to right; mean ± SEM; two-sided unpaired Student's $t$ test). **j** Representative confocal images of an explant with RFP-labelled DNPs at 3 and 5 DPI (top), and explants with RFP-labelled UNPs at 3 and 5 DPI (bottom), revealing the migration of RFP-labelled neurons from implanted cortical tissues into the brain explants. Arrowhead indicates a migrating human neuron. Scale bar, 50 μm. **k** Quantitative analysis of neuron migration from implanted cortical tissues into host brain explants ($n = 3,3,5,3,4,6,4$ biological replicates from left to right; mean ± SEM; one-way ANOVA). Field size, 0.1 mm². For **g**, **i** and **k**: ns not significant; *$P < 0.05$; **$P < 0.01$. For **c**, **d**, **f** and **h**: scale bars, 500 μm.

Gibco) and treated with 0.5 mM ethylenediaminetetraacetic acid (0,5 EDTA; Life Technologies) for 7–10 min. After aspiration of the EDTA, cells were lifted with mTeSR™Plus medium. Floating cells were collected and replated onto Geltrex-coated plates in mTeSR™Plus. On DIV0, 100% confluent hiPSCs were induced with Neural Induction Medium (NIM, Supplementary Table 1). The medium was exchanged with fresh NIM daily until DIV7. On DIV7, the cells were passaged 1:2 using 0.5 mM EDTA, replated, and cultured in NIM supplemented with 10 μM Y-27683 (Stemcell Technologies). On DIV8, the culture medium was changed to Neural Maintenance Medium (NMM, Supplementary Table 1). On DIV12, cells were passaged 1:2 again using 0.5 mM EDTA. On DIV 16, cells were passaged 1:2 using Accutase (Life Technologies). The cells were washed with DPBS first and then treated with 1 mL Accutase at 37 °C for 5–7 min. Then, an additional 3 mL DMEM/F12 medium was added (Life Technologies), followed by centrifugation for 5 min at 200×$g$. The cell pellet was resuspended with NMM supplemented with 10 μM Y-27683 and replated on Geltrex-coated plates. On DIV19, hiPSC-DNPs were passaged for either cryopreservation as cell stocks or replating for terminal maturation. For cryopreservation, DNPs were treated with 10 μM Y-27683 for three hours before dissociation with Accutase. Cell pellets were resuspended in 1 mL pre-chilled freezing medium containing 90% Fetal Bovine Serum (FBS, Gibco) and 10% DMSO (Sigma). The resuspended DNPs were transferred into cryopreservation vials and slow cooled in Freezing Containers (Thermo Scientific) at −80 °C overnight, before transfer into liquid nitrogen for long-term preservation.

**UNP differentiation.** DIV19 DNPs were cultured in NMM supplemented with a growth factor (GFs) cocktail containing 10 ng/mL fibroblast growth factor-2 (FGF-2), epidermal growth factor (EGF) and brain-derived neurotrophic factor (BDNF) (NMM+GFs, Supplementary Table 1). The culture medium was exchanged with fresh medium daily. Once the progenitors reached 100% confluency, they were passaged 1:2 using Accutase as described under DNP differentiation. The centrifuged cells were resuspended with NMM+GFs and replated. Passaging was performed approximately every 5 days until DIV 40. The DIV 40 UNPs were either cryopreserved or cultured for terminal maturation as described for DNPs.

**Maturation.** To thaw cryopreserved DNPs or UNPs, the frozen cells were placed in a 37 °C water bath. When the cells had just thawed, they were transferred to NMM and centrifuged at 200×$g$ for 4 min. The cell pellet was re-suspended in NMM with 10 μM Y-27683 and plated into one well of a Geltrex-coated 6-well plate. The cells were cultured for 2 days with a medium change after 1 day.

Then, the cells were passaged by using Accutase and replated at a concentration of 100,000/cm² on 8- or 96-well plates with Neural Terminal Medium (NTM, Supplementary Table 1) supplemented with 10 μM Y-27683. The culture medium was exchanged with fresh supplemented NTM every 3-4 days. After 10 days, mature neurons had been generated and were ready for assessment.

**Immunocytochemistry**
**Immunostaining and quantification.** DNs and UNs were cultured on Geltrex-coated 96 Well Black Polystyrene Microplates (Corning), washed with DPBS and fixed with 4% paraformaldehyde (PFA) at room temperature for 15 min. After fixation, the cells were washed 3 times with ice-cold DPBS, permeabilized by 0.1% Triton X-100 in DPBS (DPBST) for 20 min and washed again three times with DPBS. Then, the cells were blocked from non-specific binding with 10% goat serum diluted in DPBST for 60 min at room temperature. Subsequently, the neurons were incubated with primary antibodies diluted in 1% goat serum in DPBST in a humidified chamber overnight at 4 °C. Then, the neurons were washed with DPBST three times and incubated with secondary antibodies diluted in 1% goat serum in DPBST for 2 h at room temperature. Information on the antibodies used is in Supplementary Table 3. After incubation, the cells were washed three times with DPBST and once with DPBS at room temperature. Then they were mounted with Antifade Mounting Medium with DAPI (Abcam). The neurons could be examined directly or stored in the dark at 4 °C for later examination. For the quantification of cells expressing specific cortical neuronal markers, at least two images with a field of view that included over 40 cells were made. The cells were counted and the numbers were averaged for biologically independent samples.

**Real-time quantitative PCR**
Neurons were snap-frozen by putting a pellet of cells in a 15 mL centrifuge tube into powdered dry ice and stored at −80 °C. After thawing for real-time quantitative PCR (qPCR) assessment, total RNA was isolated using a Monarch® Total RNA miniprep Kit (New England BioLabs) according to the manufacturer's instructions. An additional in-tube DNase I digestion was performed to avoid the amplification of genomic DNA. RNA concentrations and quality were assessed by using a Nanodrop spectrophotometer. The RNA was reverse transcribed by using the LunaScript RT SuperMix Kit (New England BioLabs). Quantitative PCR assays were set up by loading primer mixes and cDNA into the wells of a 96-well plate in triplicate, followed by the addition of

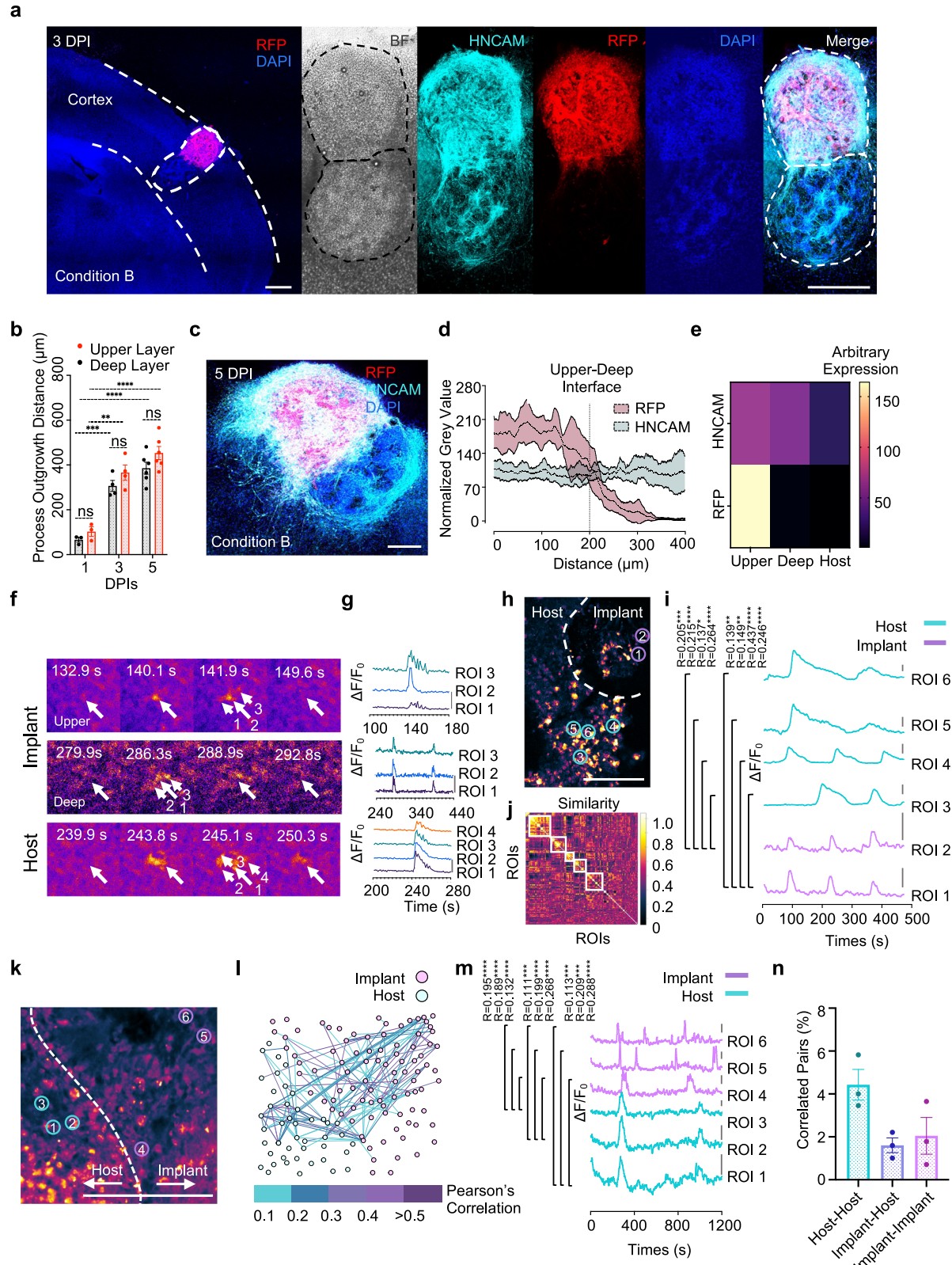

Luna Universal qPCR Master Mix (New England BioLabs). qPCR reactions were performed using the ProFlex PCR System (Applied Biosystem). Conditions were selected according to the manufacturer's suggestions (LunaScript). Relative gene expression values were determined by averaging the results of two to three technical replicates and comparing the $C_t$ values for genes of interest with those of

the control gene (18S RNA) using the $\Delta\Delta C_t$ method. Associated consumables and Primers are listed in Supplementary Tables 2 and 3.

### Droplet-printing and tissue culture

**Set up.** 3D printing was performed in a temperature- and humidity-controlled cold room. Matrigel-based bioink was printed at ~4 °C and

**Fig. 5 | Ca²⁺ activity of two-layer cortical tissue and integration with the host explant. a** Tiled fluorescence confocal image of implanted explant with RFP-labelled UNPs and unlabelled DNPs at 3 DPI. Dashed lines delineate cerebral cortex of the mouse and the implanted tissue. The human-specific neural marker HNCAM labels both layers of the implanted tissue while RFP only labels the upper layer. **b** Quantitative analysis of process outgrowth and neuron migration from the two-layered tissues into the brain explant on 1, 3 and 5 DPI ($n = 3, 3, 4, 4, 6, 6$ biological replicates from left to right; mean ± SEM; ns not significant; **$P < 0.01$, ***$P < 0.001$; ****$P < 0.0001$; two-sided unpaired Student's $t$ tests). **c** A tiled fluorescence confocal image of a two-layer implant with RFP-labelled UNPs and unlabelled DNPs at 5 DPI. **d** Fluorescence-intensity profiles showing HNCAM expression in both layers of the printed tissue, while RFP expression falls off in the deep layer ($n = 5$ biological replicates; mean ± SD). **e** Heatmap showing marker expression by grey value in the upper and deep layers of a tissue implant and in the host at 5 DPI ($n = 6$ biological replicates). **f** Video frames of Fluo-4 calcium imaging of the implanted explants reveals spontaneous calcium oscillations (coloured as fire) in individual adjacent cells at 5 DPI. **g** Single-cell Fluo-4 calcium traces. Regions of interest (ROIs) indicated

by arrows in **f**. **h** Fluo-4 calcium imaging of an explant implanted with unlabelled DNPs only at 5 DPI. Dashed line shows the interface of the implant and the host (see Supplementary Fig. 8a). **i** Correlated single-cell calcium traces between implant and the host explant, colour-coded according to their ROIs as indicated in **h** Scale bar: $\Delta F/F_0 = 0.8$. **j** Similarity matrix showing communities of neurons with correlated firing patterns. Dashed boxes show neurons that tend to activate together. Colour indicates normalised correlation. **k** Fluo-4 images of an explant implanted with RFP-labelled UNPs only at 5 DPI. The dashed line shows the interface between the implant and the host (see Supplementary Fig. 8c). **l** Correlated neuron pairs within **k**, between the implant and the host at 5 DPI by network analysis. Points representing neurons and the lines between them indicate correlated calcium signals. **m** Single-cell calcium traces, colour-coded ROIs as indicated in **k**. Scale bar: $\Delta F/F_0 = 0.05$. **n** Quantitative analysis of the correlated cell pairs within the host, within the implant and between the host and the implant. Signal pairs with Pearson's correlations over 0.1 are counted ($n = 3$ biological replicates; mean ± SEM). Slice thickness: 300 μm. For **a**, **c**, **h** and **k**: scale bars, 200 μm. For **i** and **m**. Pearson's correlation: *$P < 0.05$, **$P < 0.01$, ***$P < 0.001$ and ****$P < 0.0001$.

---

~80% humidity. Glass printing nozzles were pretreated with (3-aminopropyl)trimethoxysilane (Sigma) to provide a hydrophilic coating, which prevented the leakage of bioink and stopped oil entering the nozzles. The printing oil bath contained 2 mg/mL DPhPC (1,2-diphytanoyl-sn-glycero-3-phosphocholine, Avanti) in a mixture of undecane and silicone oil AR20 1:4 (v/v) (both from Sigma). Computer-controlled piezo-electric drivers generate the mechanical force required for droplet ejection. Compared with the previously reported setting[30,31]. Two microscopes were added to provide both front and side views, allowing the precise localisation of droplets and the alignment of multiple networks.

**Bioink preparation and droplet printing.** Cells were suspended after dissociation with Accutase. Cell number and viability were assessed after trypan blue staining (Life Technologies) by either manual counting with a hemocytometer or by using a Countess II automated cell counter (ThermoFisher). The cell suspension was then centrifuged at 200×$g$ for 5 min, followed by removal of the supernatant. Pre-thawed Matrigel at 4 °C was then added to the pellet and the cells were resuspended on ice to to generate the bioink, which was then loaded into printer nozzles at ~4 °C. Printing was performed by using custom printing-control software[30]. The printing process took 15 min to generate an $8 \times 8 \times 8$ droplet network. Patterned networks were printed either with two nozzles or by reloading a single nozzle.

**Phase transfer.** A glass cuvette containing printed networks at 4 °C was placed in a room-temperature water bath for 20 min before transfer to a tissue culture incubator at 37 °C for 1 h. Two thirds of the oil was then aspirated and replaced with a mixture of undecane and silicone oil AR20 (1: 4 v/v). The exchange process was repeated five times to dilute the lipid. Culture medium was then added drop-by-drop and exchanged four times to remove residual oil. Tissues were then transferred with a wide-bore pipette tip into 12-well or 24-well plates for subsequent culture. Compared with a previous method[31], placement of the glass cuvette in a room-temperature water bath (rather than moving to a room temperature environment) can reduce physical disturbance during solidification of the Matrigel.

**Culture of printed tissues.** Printed neural cortical tissues were cultured in NMM supplemented with FGF-2, EGF and BDNF (all at 10 ng/mL) and 100 U/mL penicillin-streptomycin (ThermoFisher) for the initial 7 days. 10 μM Y-27683 was included for the first 3 days to prevent apoptosis. Then, the cortical tissue was incubated in NTM containing 100 U/mL penicillin-streptomycin for up to 8 weeks. Half of the medium was exchanged with fresh medium every 3 days.

## Immunohistochemistry of printed tissues
**Tissue sections.** Printed tissues were fixed in 4% v/v paraformaldehyde for 1 h at room temperature and washed with DPBS three times. The fixed tissues were embedded in optimal cutting temperature compound (OCT, VWR) and sectioned using a cryostat (Leica, CM1860 UV) to generate 30-μm-thick sections on glass slides for immunostaining.

**Immunostaining.** The tissue sections were circled with a PAP pen (Merck) to create a hydrophobic barrier so that reagents could be localised. The sections were then stained according to "Immunocytochemistry" (see above). After staining, tissue sections were mounted in Mounting Medium with DAPI (Fluoroshield, Abcam) and sealed with a coverslip and nail polish for storage at 4 °C. Fluorescence visualisation was carried out with a confocal microscope and analysed with ImageJ. For quantification, three random fields were counted in each of the deep and upper layers and averaged for each biologically independent sample at 2-, 4- and 8-weeks post-printing. Three biologically independent samples were counted for each marker and timepoint.

## Implantation into mouse brain explants
Cerebral cortical tissues were printed 1 day before mouse brain slices were obtained. The next day, P8 C57BL/6 J mice were killed by cervical dislocation in accordance with the Animals Scientific Procedures Act (1986) under licence no. PP8557407. Brains were harvested and kept in ice-cold carbogen-saturated (95% O₂/ 5% CO₂) Earle's Balanced Salt Solution (EBSS, Life Technologies). Coronal slices (300 μm) were obtained by sectioning with a compresstome (Precisionary, VF-310-0Z). The brain slices (explants) were cultured on 0.4-μm Millicell-culture inserts (Merck Millipore) in six-well plates. On the same day, a biopsy punch (EMS) with 500-μm inner and 800-μm outer diameter was used to punch a lesion in the cortex of the explant. The 800-μm circle spanned most of the thickness of the cerebral cortex P7 mouse (~800–1000 μm, Allen Brain Atlas)[57]. The printed cortical tissue was then implanted into the lesion in the natural orientation (upper layer out). The implanted explants were maintained in culture medium under either nutrient condition A or nutrient condition B (Supplementary Table 1) at 37 °C under 5% CO₂. On Day 2, DAPT was added directly to the +DAPT cultures without a medium change to a final concentration of 10 nM. The culture medium was then changed every 3 days.

## Characterisation of implanted explants
**Whole-mount immunostaining.** On day 1, 3 or 5 post-implantation, brain explants were fixed in 4% v/v paraformaldehyde for 1 h at room temperature. The immunostaining protocol was similar to that in "Immunohistochemistry of printed tissues". In short, the explants were

washed with DPBS, permeabilized by using 0.5% Triton X-100 in DPBS (DPBST-0.5%) for 20 min, blocked with 10% goat serum in DPBST-0.5% for 60 min and immunostained with primary antibodies in 1% goat serum in DPBST-0.5% overnight at 4 °C. After three DPBST-0.5% washes, the explants were stained with secondary antibody (see Supplementary Table 3) in 1% goat serum in DPBST-0.5% for 2 h at room temperature, followed by DAPI staining (1 μg/mL) for 1 h and three DPBST-0.5% washes. The immunostained explants were stored in the dark in DPBS at 4 °C before imaging.

**Live/dead assay.** Implanted explants at 5 DPI were incubated with 2.5 μM calcein-AM (Cambridge Biosciences Ltd) and 5.0 μM propidium iodide (Sigma Aldrich) for 30 min before imaging. Three randomly selected cortical regions of the explants were imaged. Images were processed in ImageJ and counted manually.

**Calcium imaging.** A Fluo-4 Direct calcium assay kit (Invitrogen) was used according to the manufacturer's instructions to determine calcium activities. In short, explants on culture inserts (0.4 μm Millicell, Merck Millipore) were incubated with a mixture of BrainPhys™ Imaging Optimized Medium (Stemcell Technologies) and Fluo-4 calcium imaging reagents (1:1 v/v) for 1 h at 37 °C. After incubation, the brain explants were harvested by cutting the semi-permeable membrane from the culture insert and placing it upside down on imaging dishes (Ibidi). Spontaneous calcium fluctuations were recorded at 37 °C by fluorescence confocal microscopy (Leica SP5) at 1 frame per 1.29 s.

## Calcium imaging analysis

For the analysis of calcium imaging (Fig. 5g, i, l–n), pre-processing was performed in ImageJ. Neurons were identified as bright objects and manually selected as ROIs. The fluorescence changes of each ROI over the recording period were extracted and exported as csv files for further analysis with Excel. Subsequently, functional neuron connection analysis was performed based on the method reported by Ko et al.[61] Briefly, for each fluorescence trace, fluorescence at a given time t ($F_t$) was normalised as $\Delta F/F_0$. $F_0$ is the averaged fluorescence value for the initial 10 frames. $\Delta F = F_t - F_0$. the fluorescence trace was smoothed by averaging six-frames ($\Delta F_t/F_0 = (\Delta F_{t-2} + \Delta F_{t-1} + \Delta F_t + \Delta F_{t+1} + \Delta F_{t+2} + \Delta F_{t+3}/F_0) / 6$). Then, background fluorescence was removed from each fluorescence trace by applying a threshold which was defined as median value of the trace, with an addition of 0.02−0.05. Next, each background-corrected trace was normalised to '0–1' (with the maximum $\Delta F/F_0$ assigned as '1' and the minimum $\Delta F/F_0$ as '0'). Pearson's correlation between pairs of traces was calculated in Excel ('Data Analysis> Correlation') to generate a correlation matrix containing Pearson's correlations between all possible trace pairs. To visualise the correlations, the matrix was imported into R (Rstudio1.3.1093) by applying the plotting R package igraph (v1.3.2). We used circles to represent the location of ROIs and lines to represent the Pearson's correlation between pairs of ROIs. Following the Ho's report[61], neuron pairs with Pearson's correlation > 0.1 are defined as correlated. A blue-to-purple colour scheme was applied to the lines to represent low-to-high values of the Pearson's correlations.

To generate the similarity matrix in Fig. 5j, pre-processing, movement correction and ROI analysis of calcium imaging videos were performed automatically for all ROIs in the default settings of an open-source toolbox in MATLAB, termed NETCAL (version 8.4.1). These traces were then smoothed and a similarity matrix of all traces was generated by using default settings in NETCAL. NETCAL automatically arranged the highly correlated ROIs to the top left of the graph, and the weakly correlated ROIs to the bottom right of the graph.

## Process outgrowth and neuron migration analysis

**For printed two-layer tissues.** Sectioned two-layer tissues were analysed in ImageJ. The RFP coverage (Fig. 3f, g) was measured as the ratio of the RFP-labelled area over the total area of the deep layer, which was defined and quantified by drawing a box covering the deep-layer region and using the 'threshold' and 'measurement' tool in ImageJ. Individual migrating neurons were manually selected by using the 'cell counter' plugin in ImageJ. RFP-labelled neurons with apparent cell bodies and co-localised DAPI staining falling in the distance range of 200 to 400 μm away from the upper-layer boundary were counted as migrated neurons (Fig. 3g).

**For implanted explants.** The distance of process outgrowth in the implanted explants was measured as the farthest distance from the implant that RFP signals could be detected from single-layer implants and HNCAM signals from two-layer implants. The upper layer identity in two-layer tissues was confirmed from the extent of RFP expression. In imageJ, the fluorescence intensities were plotted with 'profile plots' and the width of the grey value decreasing phase was used to represent the outgrowth distance. Individual migrating neurons were counted by the 'cell counter' plugin in imageJ. RFP-labelled neurons with apparent cell bodies and co-localised DAPI staining falling in the distance range of 200 to 400 μm away from the implant-host boundary were counted as migrating neurons (Fig. 4j, k).

## Microscopy and image processing

Differentiated neurons were imaged using fluorescence confocal microscopes (Leica LSM780 and Leica SP5) and epi-fluorescent microscopes (Leica DMI 8 and Nikon Eclipse Ni-E). Printed cortical tissues were imaged using a fluorescence confocal microscope (Leica LSM780). Implanted explants were imaged using fluorescence confocal microscopes (Leica LSM780 and LSM980). Images were analysed by using ImageJ (version 2.1.0/1.54c). Microsoft PowerPoint (v16.66) was used for figure preparation.

## Statistics

Data in text are presented as mean ± standard error of the mean (S.E.M.). Data in figures are presented either as mean ± S.E.M. (Figs. 2d, e, 3g, i, 4g, i, k and 5b and Supplementary Fig. 7d) or mean ± standard deviation (SD) (Fig. 5d). For Figs. 2d, e and 3g, i, biological replicates n = 3. For Fig. 4k, biological replicates n ≥ 3. For Fig. 4g, biological replicates n ≥ 4. For Fig. 4i, biological replicates n ≥ 5. For Fig. 5b, d, biological replicates n ≥ 3. For Fig. 5n, biological replicates n = 3. Representative image(s) of over three experiments with similar results are shown in Figs. 1c–j, 2b, c, 3b–f, h, 4c, d, f, h, j and 5a, c, f, h, k and Supplementary Fig. 1a–e and Figs. 3a, 4a, b, 5a–d, 6a–f, 7c, e, f, and 8a, c, f. Statistical analysis was performed using GraphPad Prism 9. A detailed statistical analysis is listed in the Source Data file.

## Reporting summary

Further information on research design is available in the Nature Portfolio Reporting Summary linked to this article.

## Data availability

All data supporting the findings of this study are available within the article and its supplementary files. Any additional requests for information can be directed to, and will be fulfilled by, the corresponding authors. Source data are provided with this paper.

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

## Acknowledgements

This research was supported by a European Research Council Advanced Grant (SYNTISU) and the Oxford Martin School Programme on 3D Printing for Brain Repair. The grant-receiving authors include H.B., F.G.S., Z.M. and L.Z.

## Author contributions

Y.J., F.G.S., Z.M., L.Z. and H.B. conceived, designed and guided the project. Y.J. performed most of the experiments and analysis. Y.J., E.M. and L.Z. performed iPSCs culture. S.C. provided hiPSCs lines and advice on differentiation. Y.Z. and X.Y. assisted with confocal imaging and droplet printing. X.Y. assisted with tissue staining and Y.Z with coding and statistics. M.L. and T.S. performed animal dissections. Y.J., K.L. and D.C. performed immunostaining. Y.J. and S.B. performed preliminary explant experiments. L.C.S. assisted with qPCR. Y.J., F.G.S., Z.M., L.Z. and H.B. wrote the manuscript.

## Competing interests

The authors declare no competing interests.
