## [Peer Review File · Nature Communications]

REVIEWER COMMENTS

Reviewer #1 (Remarks to the Author):

In the manuscript, the authors claim that implantation of the printed cortical tissues into mouse brain explants resulted in substantial implant-host integration across the tissue boundaries as demonstrated by the projection of processes, the migration of neurons and the appearance of correlated Ca²⁺. Since it is very bold claim, I carefully reviewed the claim was supported by credible evidences. In particular, my primary focus was to examine the experimental results that are pertinent to the printing of two-layer cerebral cortical tissue, droplet printing technologies, and the tissue culture part. Overall conclusion is that without further strengthening the evidences the author's claims, improving the presentation methods of the experimental results, and also detailing the interpretation of the experimental data, this manuscript far in-short of acceptance to Nature Communication. Herein, I address several crucial revision points for the authors to consider. Only if those concerns are fully addressed, this manuscript can be reconsidered to be accepted to the journal.

- P3L73-74: Need to change the expression from 16x8x8 to 2 sets of 8x8x8 to avoid confusion. There is a high chance that readers might understand as another single 16x8x8 droplet network.
- P3L79: Need significant improvement on imaging editing on Extended Data Fig.1. Details are following;
 - Extended Data Fig.1: b: what is ongoing printing? Need to modify terminology for better clarity. f: Need more magnified images to identify how food dye coloured DPBS can show six-layered network. From the image provided, difficult to understand. h: Need a more magnified image. From the image, difficult to identify. Need to insert scale bar scale directly to the image for reader's comprehension.
- P3L80-82: From the images provided, I was not able to find any evidences that lipid monolayer is located at the droplet/oil interface and also contacting droplets formed droplet-interface bilayers. Since this statement is very crucial for the author's to expand their claims, sufficient evidences need to be supplemented, otherwise, whole backbone of the claims of this manuscript can be shaken.
- P3L84-86: How can you make sure that the presented droplet network can be dimensionalized as either 8x8x8 or 12x12x12? From the images, I was not able to measure them out at all
- P3L88-90: Need to change the expression from 16x8x8 to 2 sets of 8x8x8 to avoid confusion. There is a high chance that readers might understand as another single 16x8x8 droplet network.
- P17L514: Please elaborate current set-up without mentioning the term, "upgrade", which does not seem to be proper for the first-hand readers to understand in scientific community. Instead, after describing your current set-up, compare it with the previous works and describe its superiority.
- P17L536: Again, do not state " a previous protocol" and instead, fully describe your current set-up and deliberately compare yours with the previous works

Reviewer #2 (Remarks to the Author):

This manuscript by Jin et al. investigates how upper and deep-layer neurons may be 3D-printed into a cortex-like structure that resembles in vivo cortical layers. The authors pattern neurons to resemble upper and deep-layer neural progenitor cells, then print them into a 3D geometry that separates their localization. The authors then show that the printed upper and deep layer neurons retain cell identity and that the upper layer neurons project into the deep layer region. Ultimately, they investigate how the two-layer cortical structure potentially integrates into an injured brain explant by transplanting the two-layer tissue and attempting to show functional integration of the implanted tissue and the explant. The manuscript is well-written and engaging. This research expands upon this group's previously-published 3D droplet printing method, which has important implications for how the field may better model and study the cortex using in vitro cultures. Additionally, this model has potential to elevate how the field studies injury and repair to the cerebral cortex. However, this reviewer has several concerns that should be addressed.

Overall, the major concerns for this manuscript center around the fact that some of the experiments (particularly the functional ones) appear too superficial to support the claims made by the authors. Additionally, there is insufficient investigation of the biological consequences of the engraftment in the organotypic slices.

Major concerns:

1. The study currently lacks truly functional studies to show that there is interaction and functional integration between the bioprinted implant and host neurons. The correlated calcium activities leave much to be desired (see point 2).
2. The calcium signaling experiments to show association between the implant and the host brain in Figure 5 are not very convincing. If the authors want to claim that the explant and bioprinted cells are functionally integrated, this will require more in-depth characterization.
3. If the goal of the paper is to show that this bioprinting system could be used to repair damage to the brain, more than just cell interactions are required. By what mechanism do the authors believe the implant is potentially helpful? By functional integration into circuits? By secretion of trophic factors? This remains a big unknown.
4. To show the accuracy of generating upper and deep layer neurons, it would be easy and far more quantitative to perform RNA-sequencing to see how well the deep and upper layer neurons recapitulate in vivo development and neuronal transcriptomes.
5. The authors should make it clear in the text that in Figure 5, the implants are either deep layer or upper layer neurons exclusively (I believe it is only mentioned in the figure legend). More importantly though, why is this the case? Why analyze them separately when the novelty of this approach is the fact that there are both types of neurons, which may lead to better recovery outcomes? If they are kept

separate, this needs to be abundantly more clear in the figure, itself, instead of switching back and forth arbitrarily.

6. The manuscript overall struggles to make a compelling argument about the translatability of this approach. Engrafting a ball of neurons into an injured brain makes many assumptions about how this will actually be beneficial. The authors need to more clearly explain how they see the potential use of this approach and why it could possibly be effective.

7. What about glial cells or reactive glia around the wound site in the mouse brain explants? The authors should investigate or include discussion on how the bioprinted implant may affect other cell types in the surrounding parenchyma.

Minor concerns:

1. This reviewer is concerned about the print fidelity of this bioprinting system. In Figure 3H, it appears that there is a large difference in size between the upper and deep layers between samples/prints. Could the authors show metrics that they use to ensure print reproducibility?

2. Figure 3H shows the printed cortical tissues stained with various markers for upper and deep layer neurons. Would it be possible to perform double-stains on these sections to better understand if the same cells that express one marker also express another? Additionally, is there a better marker for deep layer neurons than CTIP2 (such as TBR1)? Because the expression of CTIP2 fades over time, it is important to show that the printed deep layer neurons are maintaining their identity.

3. Please expand on the cell lines used within the figure legends. Were 3 iPSC lines used for each experiment? This is largely unclear from experiment to experiment.

4. Figure 4K was not cited within the text, please correct this.

5. There are multiple inconsistent measurements of Deep and Upper layer explants, such as in Figure 4J and 4K. Could the authors include immunostaining images for deep layer neurons at 3DPI to make them comparable to the upper layer neurons, which were imaged at 3DPI and 5DPI?

Reviewer #3 (Remarks to the Author):

Summary

In the submitted manuscript, the authors present a composite cell-ECM bioprinting technique to generate layered structures aimed at recapitulating the cortical layers in the brain. The authors are commended for a well-planned experimental design and clear presentation of the results in the manuscript. The main finding of this paper was that a combination of hiPSC differentiation protocols and bioprinting enabled developing a two-layer composite comprised of upper neural progenitor and deep neural progenitor cells. The authors demonstrate the functionality of this system by providing robust

IHC and qRT-PCR characterization of the composite from in vitro cultures. The final study was an ex vivo “implantation” of the composite in a “lesioned” cortex. The main critiques for this study include the overstatement in modeling traumatic brain injury (TBI), extrapolation of an ex vivo model to in vivo assumptions, and lack of electrophysiological characterization of this system. Therefore, the conclusions are only supported by IHC/transcriptomic data rather than expansive functional assessments. Overall, the manuscript presents potentially unique contributions to the neural regenerative medicine and organoid fields.

Concerns/Comments

1. Overstatement for findings and focus on TBI:

a. The title of the manuscript, “Functional Integration of 3D-Printed Cerebral Cortical Tissue into a Brain Lesion” implies that an in vivo study was performed. The model used was an ex vivo cortical slice that then was “lesioned”. This model is insufficient to model the complexity of a TBI. Therefore, the title should be adjusted to reflect the study – Ex vivo should be stated in the title.

b. The introduction focuses largely on TBI – however, as stated above, this approach and model does not model specifically TBI. Therefore, the introduction could be more expansive and inclusive for additional neurological conditions such as stroke and surgical resection for cancer and epilepsy.

2. Functional assessment: The authors acknowledge that the short timepoint post-implantation (5days) limits the functional assays to demonstrate integration of the “transplant” cells with host cortical tissue/cellular networks. The calcium imaging is a good first step, yet, the impact of this approach would be bolstered by longer culture times and electrophysiological assessments.

3. Demonstrate the necessity for the two layers for cortical development. The in vitro characterization of the two layer composite system consisting of the upper and deep layer neural progenitors was well supported. However, a comparison to single cell type composites (i.e., upper+upper, or deep+deep) would have demonstrated potential synergy for subsequent maturation and differentiation of such complex cortical architecture.

4. Limitation for Matrigel-based encapsulation/printing technique. The use of Matrigel ECM for bioprinting is limited due to complications with production, reproducibility, and availability. This point should be discussed and addressed as a limit for moving this approach forward for further development.

REVIEWER COMMENTS

Reviewer #1 (Remarks to the Author):

In the manuscript, the authors claim that implantation of the printed cortical tissues into mouse brain explants resulted in substantial implant-host integration across the tissue boundaries as demonstrated by the projection of processes, the migration of neurons and the appearance of correlated Ca²⁺. Since it is very bold claim, I carefully reviewed the claim was supported by credible evidences. In particular, my primary focus was to examine the experimental results that are pertinent to the printing of two-layer cerebral cortical tissue, droplet printing technologies, and the tissue culture part. Overall conclusion is that without further strengthening the evidences the author's claims, improving the presentation methods of the experimental results, and also detailing the interpretation of the experimental data, this manuscript far in-short of acceptance to Nature Communication. Herein, I address several crucial revision points for the authors to consider. Only if those concerns are fully addressed, this manuscript can be reconsidered to be accepted to the journal.

- P3L73-74: Need to change the expression from 16x8x8 to 2 sets of 8x8x8 to avoid confusion. There is a high chance that readers might understand as another single 16x8x8 droplet network.

We agree that the expression is confusing. We have made the suggested change from '16x8x8' to '2 sets of 8x8x8'.

At Lines 89, 208 and 894: "16x8x8" was changed to "two sets of 8x8x8...."

- P3L79: Need significant improvement on imaging editing on Extended Data Fig.1. Details are following;
- Extended Data Fig.1: b: what is ongoing printing? Need to modify terminology for better clarity. f: Need more magnified images to identify how food dye coloured DPBS can show six-layered network. From the image provided, difficult to understand. h: Need a more magnified image. From the image, difficult to identify. Need to insert scale bar scale directly to the image for reader's comprehension.

We have changed the figure legend of Extended Data Fig 1b from 'Side-view image of on-going printing.' to 'Side-view during the printing process.' In addition, we have included a magnified image for the six-layer network made containing food dyes in DPBS (Response Figure 1; also as Extended Data Fig 1f). Extended Data Fig 1h has also been replaced with a more magnified image containing a scale bar.

Response Figure 1 (Extended Data Fig 1). **f.** Six-layered network of droplets containing food-dye-coloured DPBS. Scale bar, 1000 μm . **g.** View of the six-layered network in 'f' alongside a ten-pence coin. **h.** View of the droplet network in Fig. 1f. The network was printed as a mirror image of 'OXF' for imaging with an inverted microscope. For 'g' and 'h': scale bar, 500 μm .

Dr Alessandro Alcinesio and colleagues from our group revealed the structure at the interface of two networks (Response Figure 2)¹. DIBs form between droplets at the interface and prevent the exchange of contents between droplets¹.

Response Figure 2 (Figure 2c of cited study¹). Confocal microscopy image of the interface between two assembled droplet building blocks containing fluorescent dyes with high affinities for lipid bilayers.

- P3L80-82: From the images provided, I was not able to find any evidences that lipid monolayer is located at the droplet/oil interface and also contacting droplets formed droplet-interface bilayers. Since this statement is very crucial for the author's to expand their claims, sufficient evidences need to be supplemented, otherwise, whole backbone of the claims of this -- manuscript can be shaken.

To demonstrate the formation of droplet-interface bilayers, Dr Linna Zhou and colleagues from our group used Texas-red-labeled lipids to visualize the DIBs in a printed network containing Matrigel and cells (Response Figure 3) ².

Response Figure 3 (Figure 1c of the cited study ²). Left: bright-field image of part of a printed droplet network. Droplets contain Matrigel and HepG2 cells at $3 \times 10^7 \text{ mL}^{-1}$. Right: Texas-red-labeled lipids reveal DIBs in a printed network.

One of the important features of DIBs is the contact angle between droplets. Our previous publication by Dr Alessandro Alcinesio and colleagues revealed that the equilibrium contact angle of DIBs between a pair of droplets is a key parameter of tessellation and precise positioning of droplets. The printed network form regular hexagonal close-packed lattices with least defect when the contact angle reached a geometrically-derived critical angle of 35.3° . A hexagonal droplet with DIB contact angle of $\sim 35^\circ$ in network is shown below (Response Figure 4) ³.

Response Figure 4 (Figure 5g–i of the cited study ³). Droplets sectioned through the z-axis from bottom to top, with (inset) computer models of trapezo-rhombic dodecahedra showing 2D sections of three-fold (g, i) and six-fold (h) symmetry.

- P3L84-86: How can you make sure that the presented droplet network can be dimensionalized as either 8x8x8 or 12x12x12? From the images, I was not able to measure them out at all

Our printing process is controlled by a custom computer program that allows us to apply specific printing maps for desired droplet network structures. In this study, we created 8x8x8 and 12x12x12 printing maps (Response Figure 5) which were uploaded into the program and used to generate the corresponding network structures.

Response Figure 5. Printing maps for 8X8X8 (left) or 12x12x12 (right) networks.

We understand that the printing process contains defects. In particular, droplets occasionally roll over from the top layer. Dr Alessandro Alcinesio from our group worked on the improvement of network packing and reported an optimal DIB contact angle leading to the least defect (Response Figure 6)³.

Response Figure 6 (Figure 4a the cited study³). Overlaid confocal microscopy images (co-registered) (n = 5) of the first layer of 3D-printed 7x8 droplet networks, which had a hexagonal

packing fraction of 0.50 ± 0.07 (contact angle $\theta_{\text{DIB}} = 36.3^\circ$, volume fraction of silicone oil $\phi_{\text{SIL}} = 0.55$, mole fraction of POPC $x_{\text{POPC}} = 0.13$).

- P3L88-90: Need to change the expression from 16x8x8 to 2 sets of 8x8x8 to avoid confusion. There is a high chance that readers might understand as another single 16x8x8 droplet network.

We have modified the 16x8x8 to 2 sets of 8x8x8.

At Lines 89, 208 and 894: "16x8x8" was changed to "two sets of 8x8x8...."

- P17L514: Please elaborate current set-up without mentioning the term, "upgrade", which does not seem to be proper for the first-hand readers to understand in scientific community. Instead, after describing your current set-up, compare it with the previous works and describe its superiority.

We have included further description of our current droplet printing technology and explained the improvements over the previous version.

From P18L536 to P18L539, we modified to (highlighted sections below and in the main text are the modified content):

“Set up. 3D printing was performed in a temperature- and humidity-controlled cold room. Matrigel-based bioink was printed at $\sim 4^\circ\text{C}$ and $\sim 80\%$ humidity. Glass printing nozzles were pretreated with (3-aminopropyl)trimethoxysilane (Sigma) to provide a hydrophilic coating, which prevented the leakage of bioink and stopped oil entering the nozzles. The printing oil bath contained 2 mg/mL DPhPC (1,2-diphytanoyl-sn-glycero-3-phosphocholine, Avanti) in a mixture of undecane and silicone oil AR20 1:4 (v/v) (both from Sigma). Computer-controlled piezo-electric drivers generate the mechanical force required for droplet ejection. Compared with the previously reported setting^{4,5}. Two microscopes were added to provide both front and side views, allowing the precise localisation of droplets and the alignment of multiple networks. The current printing setup associated consumables are listed in Supplementary Table 2.”

- P18L536: Again, do not state “ a previous protocol” and instead, fully describe your current set-up and deliberately compare yours with the previous works

We have included a more detailed introduction to our current printing setup and explained the settings and functions that were modified and improved over the previous version.

From P19L558 to P19L562, we modified to:

“Phase transfer. A glass cuvette containing printed networks at 4°C was placed in a room-temperature water bath for 20 min before transfer to a tissue culture incubator at 37°C for 1 h. Two thirds of the oil was then aspirated and replaced with a mixture of undecane and silicone oil AR20 (1: 4 v/v). The exchange process was repeated five times to dilute the lipid. Culture medium was then added drop-by-drop and exchanged four times to remove residual oil. Tissues were then transferred by wide-bore pipette tips into 12-well or 24-well plates for subsequent culture. Compared with the previous method ⁵, placement of the glass cuvette in a room-temperature water bath (rather than moving to a room temperature environment) can reduce physical disturbance during early solidification of the Matrigel.”

Reviewer #2 (Remarks to the Author):

This manuscript by Jin et al. investigates how upper and deep-layer neurons may be 3D-printed into a cortex-like structure that resembles *in vivo* cortical layers. The authors pattern neurons to resemble upper and deep-layer neural progenitor cells, then print them into a 3D geometry that separates their localization. The authors then show that the printed upper and deep layer neurons retain cell identity and that the upper layer neurons project into the deep layer region. Ultimately, they investigate how the two-layer cortical structure potentially integrates into an injured brain explant by transplanting the two-layer tissue and attempting to show functional integration of the implanted tissue and the explant. The manuscript is well-written and engaging. This research expands upon this group's previously-published 3D droplet printing method, which has important implications for how the field may better model and study the cortex using *in vitro* cultures. Additionally, this model has potential to elevate how the field studies injury and repair to the cerebral cortex. However, this reviewer has several concerns that should be addressed.

Overall, the major concerns for this manuscript center around the fact that some of the experiments (particularly the functional ones) appear too superficial to support the claims made by the authors. Additionally, there is insufficient investigation of the biological consequences of the engraftment in the organotypic slices.

Major concerns:

1. The study currently lacks truly functional studies to show that there is interaction and functional integration between the bioprinted implant and host neurons. The correlated calcium activities leave much to be desired (see point 2).

Our response to this point is incorporated into the response to point 2.

2. The calcium signaling experiments to show association between the implant and the host brain in Figure 5 are not very convincing. If the authors want to claim that the explant and bioprinted cells are functionally integrated, this will require more in-depth characterization.

We have adjusted our statement to accentuate the finding of correlated Ca^{2+} oscillations, without claiming full functional integration. These changes are highlighted in the main text at P1L31, P3L63, P12L347, 350 and P14L410, 411, where 'Functional integration' was changed to 'correlated calcium oscillation' or 'correlated Ca^{2+} activity'.

In the current study, we applied Fluo-4 (a fluorescent calcium indicator) to investigate neuronal activity in the implanted tissue and the host brain slice and found spontaneous calcium ion oscillations in both tissues. In a next step, we recorded spontaneous calcium ion oscillations at the interface between the implanted tissue (upper or deep layer) and the host brain slice. We found that multiple neurons separated by the interface exhibited correlated Ca^{2+} oscillations (correlation factor $R > 0.1$). Ko *et al.* demonstrated that neurons with correlation factor $R > 0.1$ are

likely to have formed connections, which was proved by patch-clamp recording these neurons in mouse cortex (Response Figure 7; also as Figure 3 a and b of the cited study) ⁶.

Response Figure 7 (Figure 3 of the cited study) ⁶. **a**, An example of a triplet of neurons targeted for whole-cell recording in vitro, with associated in vivo calcium responses to the natural movie (average of six repetitions) and spike rate correlation values. Neuron 1 and 2 showed correlated firing (signal correlation 0.31), whereas other pairs did not. **b**, Triple recordings from the same neurons reveal the pattern of connections: neurons 1 and 2 were bidirectionally connected, whereas neuron 3 provided input to neuron 1. Dashed lines indicate timing of presynaptic spikes.

(In this referenced study, the spontaneous activity of the exemplified three neurons were recorded through Ca^{2+} imaging in living rodent brain while providing visual stimulation with a movie. Subsequently, the same three neurons were stimulated and recorded by path-clamping to confirm their connectivity)

3. If the goal of the paper is to show that this bioprinting system could be used to repair damage to the brain, more than just cell interactions are required. By what mechanism do the authors believe the implant is potentially helpful? By functional integration into circuits? By secretion of trophic factors? This remains a big unknown.

Three-dimensional printed cortical tissues might benefit brain repair in several ways.

Firstly, adult brain lacks regenerative capacity. Although cells with stem-cell-like properties can occur throughout the adult central nervous system, they normally give rise to neurons in a few restricted areas ⁷. The implantation of 3D printed cortical tissue can therefore serve as a viable option to replace lost cells and damaged tissue caused by traumatic brain injury (TBI) or other conditions.

Secondly, the outgrowth of processes and migration of cells from the implant into the host tissue is indicative of integration. This signifies the structural integration of the implant into the host circuits, providing a potential avenue for functional recovery.

Lastly, the presence of correlated calcium signals suggests an early stage of communication to the brain circuits. However, long-term integration with circuit formation requires further validation through behavioural experiments on living animals, which is beyond the scope of the current study.

4. To show the accuracy of generating upper and deep layer neurons, it would be easy and far more quantitative to perform RNA-sequencing to see how well the deep and upper layer neurons recapitulate *in vivo* development and neuronal transcriptomes.

The human iPSCs and neuron differentiation protocols in our study have been reported by Dr Sally Cowley, Oxford Stem Cell Institute ⁸. These reported protocols are very similar to the widely used Shi Protocol. According to Shi's protocol, early neurons (<Day35) exhibit a deep-layer phenotype, including the expression of TBR1 and CTIP2 ⁹. For upper-layer neuron differentiation, we followed the protocol published by Boissart *et al.* The Boissart protocol reported a significantly increased upper-layer specific mRNA expression (CUX1, CUX2, BRN1 and BRN2) and protein marker expression (CUX1 and CUX2) on the neurons with age >Day 50 ¹⁰. In the current study, we applied immunostaining to confirm that >70% upper-layer neuron (DIV 50+) express upper-layer markers, such as CUX1, BRN2 and SATB2, and <16% upper-layer neuron express deep-layer marker such as CTIP2. By contrast, immunostaining revealed almost no expression of CUX1, BRN2 and SATB2 in deep-layer neurons (DIV 29) but strong expression of CTIP2. We also applied RT-qPCR, which revealed a significant increase in the mRNA levels of upper-layer markers CUX1 and CUX2 in both upper-layer neurons and upper-layer neural progenitors.

RNA sequencing of iPSC-cortical neurons has been reported in literature and provided further information for science community, particularly a comparison between *in vitro* differentiated cortical neurons and their counterparts *in vivo* during brain development. For example, a recently published study, applying LDN193198 and SB431542 for neural induction similar as our study, reported a comprehensive transcriptomic data covering the differentiating neurons at day 2, 4, 6, 9, 15, 21, 49, 63 and 77 ¹¹. The study illustrated that about 23% of the RNA from *in vitro* neuronal cultures with over 8 weeks differentiation reflected signatures of adult cortical neurons. This percentage increased to 48% if the neurons are cultured in a more natural environment, such as co-cultured with astrocytes.

5. The authors should make it clear in the text that in Figure 5, the implants are either deep layer or upper layer neurons exclusively (I believe it is only mentioned in the figure legend). More importantly though, why is this the case? Why analyze them separately when the novelty of this approach is the fact that there are both types of neurons, which may lead to better recovery outcomes? If they are kept separate, this needs to be abundantly more clear in the figure, itself, instead of switching back and forth arbitrarily.

We have edited the manuscript to clearly state that the implants are either deep or upper layer exclusively in Figure 5 panel h-n. We initially focused on the 3D printed deep- or upper-layer

only cortical tissues to ensure the success of implantation in brain explants. Then we implanted two-layer cortical tissues in brain explants to match the mouse cortex layering structure. Here, we performed additional calcium imaging analysis on the implanted two-layer tissue and revealed the correlated calcium ion signals between the implant and host (Response Figure 8; also as Extended Data Fig 7f-h).

Response Figure 8 (Extended Data Fig 7). **f.** Confocal images of an explant implanted with two-layer tissue containing RFP-labelled UNPs and unlabelled DNP at 5 DPI. The dashed lines show the interfaces between the upper layer of the implant, the deep layer of the implant and the host. The solid boxes show a magnified view of RFP-labelled upper layer of the implant with a strong contrast, demonstrating migrating RFP-labelled neurons. Scale bar in the box, 50 μm . **g.** Single-cell calcium traces, colour-coded ROIs as indicated in 'f'. Scale bar: $\Delta F/F_0 = 0.05$; R: Pearson's correlation value; and P-value of Pearson's correlation: ****, $P < 0.0001$. **h.** Correlated neuron pairs within 'f', among the upper layer of the implant, the deep layer of the implant and the host at 5 DPI by network analysis in conventional (left) and circular layout (right). Points representing neurons and the lines between them indicate correlated calcium signals. For all panels: scale bar, 200 μm .

P12L344: "To seek correlations of calcium oscillations between implant and host, we recorded Ca^{2+} signals at the implant/explant interface (Fig. 5h and Supplementary Video 2)."

was changed to

"To seek correlations of calcium oscillations between implant and host, we recorded Ca^{2+} signals at the **deep-layer only** implant/ explant interface (Fig. 5h and Supplementary Video 2)."

P12L350: "Further Ca^{2+} imaging (Fig. 5k and Extended Data Fig. 7c,d) and correlated network assessment (Fig. 5m and Extended Data Fig. 7e) on 5 DPI implanted explants showed correlated cell pairs within the implant and the host, and across the implant/ explant boundary."

was changed to

“Further Ca^{2+} imaging and correlated network assessment on implanted 5 DPI upper-layer only tissue (Fig. 5k,m and Extended Data Fig. 7c-e) and implanted two-layer tissue (Extended Data Fig. 7 f-h) showed correlated cell pairs within the implant and the host, and across the upper/deep and implant/host boundaries.”

6. The manuscript overall struggles to make a compelling argument about the translatability of this approach. Engrafting a ball of neurons into an injured brain makes many assumptions about how this will actually be beneficial. The authors need to more clearly explain how they see the potential use of this approach and why it could possibly be effective.

We have improved our discussion section to better explain the potential benefits of 3D fabricated neural tissue by comparison with other methods for tissue repair.

Most adult brain regions cannot regenerate post injury and no effective treatment for TBI has been reported¹². Current cell replacement therapies, such as the injection of dissociated cells, do not restore the normal brain cellular profile and tissue architecture in the brain^{13,14}. The implantation organoids into rat brain provided multiple types of brain cells (neuronal progenitors, mature neurons, astrocytes etc.) and produced functional connectivity between implant and host. However, while immunostaining for SATB2 and CTIP2 revealed the presence of cortical layer subtypes in the implanted organoids, no obvious anatomical lamination was observed¹⁵. Three-dimensional printing can fabricate tissue with a pre-designed architecture, resembling the brain anatomy essential for higher cognition. In our work, 3D printed cortical tissues implanted into brain explants show process outgrowth and cell migration into the host brain, illustrating structural integration between implant and host. The structural integration might serve as a critical early step in the restoration of injured brain cortex. In addition, the implanted cortical tissue demonstrated correlated Ca^{2+} oscillations with host. Therefore, the 3D printing technique provides an initial proof-of-concept for an approach to treat diseases currently without a cure.

P14L405: “Particularly in the case of a large lesion, the implantation of replacement tissue with matched 3D shape and cellular architecture will provide a precise treatment.”

was modified to

“Particularly in the case of a large lesion, the implantation of replacement tissue with matched 3D shape and cellular architecture, resembling the essential brain anatomy for higher cognition, will provide a precise treatment.”

7. What about glial cells or reactive glia around the wound site in the mouse brain explants? The authors should investigate or include discussion on how the bioprinted implant may affect other cell types in the surrounding parenchyma.

We performed immunostaining with biomarkers GFAP and S100beta for astrocytes on implanted brain explants. We found astrocytes in the implanted cortical tissues (Response Figure 9). Printed cortical tissues after 2 WPP *in vitro* culture (without implantation) do not contain GFAP positive astrocytes. Therefore, the astrocytes in the implant originate from the host brain explant. The migration of astrocytes is expected as a part of host brain's immune response. Autologous transplantation of iPSC-neurons have been reported to elicit only a minimal immune response in the nonhuman primate brains, compared to allografts¹⁶. In the possible clinical scenario, the use of human iPSCs reprogrammed from patient's own cells could potentially relieve long-term rejection effect¹⁷.

Response Figure 9 a. Tiled z-stack fluorescence confocal image of an implanted explant immunostained for the astrocyte biomarker GFAP. The implant contains RFP-labelled UNPs and unlabelled DNPs at 5 DPI. White dashed lines delineate the implanted tissue. **b.** Magnified tiled z-stack fluorescence confocal images at the interface of the implant and brain explant, as indicated by the red dashed box in panel 'a'. RFP labels UNPs and DAPI labels UNPs, DNPs and host cells. GFAP labels astrocytes in brain explant and astrocytes that have penetrated into the implant. **c.** Tiled z-stack fluorescence confocal images of another implanted explant with UNPs

only at 5 DPIs immunostained for the astrocyte biomarker GFAP and S100beta. For 'a', 'b' and 'c': scale bars, 200 μ m.

Minor concerns:

1. This reviewer is concerned about the print fidelity of this bioprinting system. In Figure 3H, it appears that there is a large difference in size between the upper and deep layers between samples/prints. Could the authors show metrics that they use to ensure print reproducibility?

We include a bar graph below to summarise differences between the upper and deep layers in printed tissues. Several factors can cause the variation of size between the upper and deep layers. First, printing defects (droplet packing defects, variation in droplet size etc.) can potentially lead to differences of printed networks. In addition, during post printing culture, progenitors in the deep and upper layers may proliferate at different rates. Further, we sectioned printed tissue into 30 μ m slices and the section angle can also contribute to the apparent geometry of the tissue.

Response Figure 10. A bar graph demonstrates the length ratio of upper or deep layer in vitro cultured tissue at 2, 4 and 8 WPP (n=3; ns=no significance).

2. Figure 3H shows the printed cortical tissues stained with various markers for upper and deep layer neurons. Would it be possible to perform double-stains on these sections to better understand if the same cells that express one marker also express another? Additionally, is there a better marker for deep layer neurons than CTIP2 (such as TBR1)? Because the expression of CTIP2 fades over time, it is important to show that the printed deep layer neurons are maintaining their identity.

Our layer-specific antibodies (CUX1, SATB2 & TBR1) are widely used in many publications and all originate from rabbits. Therefore, double-staining with these antibodies will cause cross-

reaction. We have attempted to use primary antibodies from alternative host species. However, they did not perform as well as the current options, and resulted in unspecific bindings and a strong background. The quality of other deep-layer marker immunostainings, like TBR1, remains inferior to that of CTIP2, leading to some nonspecific binding within the cytoplasm which affects quantification.

3. Please expand on the cell lines used within the figure legends. Were 3 iPSC lines used for each experiment? This is largely unclear from experiment to experiment.

We used three lines in our study, unlabelled AH016-3 hiPSCs, RFP-labelled AH016-3 hiPSCs and GFP-labelled AH016-3. The original AH016-3 hiPSC line was reprogrammed from the biopsy acquired from a disease-free, 80 years-old, male donor ¹⁸.

We modified all figure legends to incorporate the cell line information for each panel.

4. Figure 4K was not cited within the text, please correct this.

We have cited the Figure 4k in the main text at P11L310 and move the citation of Figure 4j to P11L316 (see below).

“Under high magnification, we identified individual neurons migrating across the implant-host boundary (Fig. 4k). Quantitative analysis was conducted by counting the number of RFP-labelled neurons in an area between 200 and 400 μm away from the implant. The analysis showed 20 ± 2 , 17 ± 3 and 18 ± 3 RFP-labelled neurons/ 0.1mm^2 had migrated into the host brain explant at 5 DPIs from implants composed of UNs, DNs, and 14 days pre-cultured DNs respectively. The migration of DNs (0.7 ± 0.4 , 13 ± 3 , 17 ± 3 neurons/ 0.1mm^2 at 1, 3 and 5 DPI) and UNs (1.4 ± 0.4 , 12 ± 4 , 20 ± 2 neurons/ 0.1mm^2 at 1, 3 and 5 DPI) were observed over three time points (Fig. 4j).

5. There are multiple inconsistent measurements of Deep and Upper layer explants, such as in Figure 4J and 4K. Could the authors include immunostaining images for deep layer neurons at 3DPI to make them comparable to the upper layer neurons, which were imaged at 3DPI and 5DPI?

We have included a neuron migration assessment of 1&3 DPI deep-layer implanted brain explant (Response Figure 11; also as Figure 4J and K). We have also adjusted the contrast of the image in Figure 4J (bottom) to match the other images.

Response Figure 11 (Figure 4). **j.** Representative confocal images of an explant with DNs at 3 DPI and 5 DPI (top), and explants with UNs at 3 DPI and 5 DPI (bottom), revealing the migration of RFP-labelled neurons from implanted cortical tissues into the brain explants. Arrow indicates a migrating human neuron. Scale bar, 50 μ m. **k.** Quantitative analysis of neuron migration from implanted cortical tissues into host brain explants ($n \geq 3$; one-way ANOVA). Field size, 0.1mm². For 'k': ns = not significant; *, $P < 0.05$; **, $P < 0.01$.

Reviewer #3 (Remarks to the Author):

Summary

In the submitted manuscript, the authors present a composite cell-ECM bioprinting technique to generate layered structures aimed at recapitulating the cortical layers in the brain. The authors are commended for a well-planned experimental design and clear presentation of the results in the manuscript. The main finding of this paper was that a combination of hiPSC differentiation protocols and bioprinting enabled developing a two-layer composite comprised of upper neural progenitor and deep neural progenitor cells. The authors demonstrate the functionality of this system by providing robust IHC and qRT-PCR characterization of the composite from in vitro cultures. The final study was an ex vivo “implantation” of the composite in a “lesioned” cortex. The main critiques for this study include the overstatement in modeling traumatic brain injury (TBI), extrapolation of an ex vivo model to in vivo assumptions, and lack of electrophysiological characterization of this system. Therefore, the conclusions are only supported by IHC/transcriptomic data rather than expansive functional assessments. Overall, the manuscript presents potentially unique contributions to the neural regenerative medicine and organoid fields.

Concerns/Comments

1. Overstatement for findings and focus on TBI:

a. The title of the manuscript, “Functional Integration of 3D-Printed Cerebral Cortical Tissue into a Brain Lesion” implies that an in vivo study was performed. The model used was an ex vivo cortical slice that then was “lesioned”. This model is insufficient to model the complexity of a TBI. Therefore, the title should be adjusted to reflect the study – Ex vivo should be stated in the title.

We have changed the title to “Integration of 3D-Printed Cerebral Cortical Tissue into a lesioned Brain Slice”

b. The introduction focuses largely on TBI – however, as stated above, this approach and model does not model specifically TBI. Therefore, the introduction could be more expansive and inclusive for additional neurological conditions such as stroke and surgical resection for cancer and epilepsy.

We have modified the introduction along the lines suggest by the reviewer. In the modified introduction, we discuss the effects of brain injuries, including traumatic brain injury (TBI), stroke, surgical resection for cancer, and epilepsy. Individuals with these injuries may experience a variety of symptoms, including cognitive dysfunction, motor impairment, and difficulty communicating.

At P2L44, we modified our introduction :” Brain injuries, which include traumatic brain injury (TBI)⁵, stroke¹⁹, surgical resection for cancer²⁰, and epilepsy²¹, can result in significant damage

to the cerebral cortex, leading to a range of symptoms for individuals, including cognitive dysfunction²²⁻²⁵, motor impairment²⁶⁻²⁹ and difficulty communicating³⁰⁻³³, and burdens for society. For example, it was reported in 2018 that 69 million people globally suffer from TBI and 4.8 million of these cases are severe or fatal³⁴⁻³⁷.

2. Functional assessment: The authors acknowledge that the short timepoint post-implantation (5days) limits the functional assays to demonstrate integration of the “transplant” cells with host cortical tissue/cellular networks. The calcium imaging is a good first step, yet, the impact of this approach would be bolstered by longer culture times and electrophysiological assessments.

We have adjusted our statement to accentuate the finding of correlated Ca^{2+} oscillations, without claiming full functional integration. These changes are highlighted in the main text at P1L31, P3L63, P12L347, 350 and P14L410, 411, where ‘Functional integration’ was changed to ‘correlated calcium oscillation’ or ‘correlated Ca^{2+} activity’.

In the current study, we applied Fluo-4 (a fluorescent calcium indicator) to investigate neuronal activity in the implanted tissue and the host brain slice and found spontaneous calcium ion oscillations in both tissues. In a next step, we recorded spontaneous calcium ion oscillations at the interface between the implanted tissue (upper or deep layer) and the host brain slice. We found that multiple neurons separated by the interface exhibited correlated Ca^{2+} oscillations (correlation factor $R > 0.1$). Ko *et al.* demonstrated that neurons with correlation factor $R > 0.1$ are likely to have formed connections, which was proved by patch-clamp recording these neurons in mouse cortex (Response Figure 12; also as Figure 3 a and b of the cited study)⁶.

Response Figure 12 (Figure 3 of the cited study)⁶. **a**, An example of a triplet of neurons targeted for whole-cell recording in vitro, with associated in vivo calcium responses to the natural movie (average of six repetitions) and spike rate correlation values. Neuron 1 and 2 showed correlated firing (signal correlation 0.31), whereas other pairs did not. **b**, Triple recordings from the same neurons reveal the pattern of connections: neurons 1 and 2 were bidirectionally connected, whereas neuron 3 provided input to neuron 1. Dashed lines indicate timing of presynaptic spikes.

(In this referenced study, the spontaneous activity of the exemplified three neurons were recorded through Ca^{2+} imaging in living rodent brain while providing visual stimulation with a movie. Subsequently, the same three neurons were stimulated and recorded by path-clamping to confirm their connectivity)

3. Demonstrate the necessity for the two layers for cortical development. The *in vitro* characterization of the two layer composite system consisting of the upper and deep layer neural progenitors was well supported. However, a comparison to single cell type composites (i.e., upper+upper, or deep+deep) would have demonstrated potential synergy for subsequent maturation and differentiation of such complex cortical architecture.

We have compared the expression pattern of upper- and deep-layer markers in 8 WPP *in vitro* cultured upper-layer only tissue and deep-layer only tissue (Response Figure 13, also as Extended Data Fig 5b). We found upper-only and deep-only tissue demonstrated a similar percentage of CUX1 and CTIP2-expressing cells to the upper and deep layer of two-layer tissue. We did not observe a geographical enrichment of CUX1- and CTIP2-expressing cells. This shows single-layer tissues do not differentiate and segregate into a multi-layer structure within 8 weeks.

Response Figure 13 (Extended Data Fig 5b) Fluorescence images of sectioned 8 WPP unlabelled deep-layer only cortical tissues (top) and RFP-labelled upper-layer only cortical tissues (bottom) showing the expression of the deep-layer markers (CTIP2 & TBR1) and upper-layer marker (CUX1).

At P7L204, “Upper-layer cortical tissue at 8WPP, in contrast, demonstrated CUX1-expressing UNs and sparse CTIP2/TBR1-expressing DNs (Extended Data Fig. 5b).” was added.

4. Limitation for Matrigel-based encapsulation/printing technique. The use of Matrigel ECM for

bioprinting is limited due to complications with production, reproducibility, and availability. This point should be discussed and addressed as a limit for moving this approach forward for further development.

Matrigel indeed could lead to potential risk in a clinical scenario. We have included a more detailed discussion of the limitations of Matrigel, such as batch-to-batch variation, limited supply and safety concerns over its tumor-derived origin. Hydrogels isolated from the ECM of specific healthy tissues have been reported to be an alternative to Matrigel to produce organoids³⁸. Although the composition of these hydrogels could still vary from batch to batch.

To overcome the limitations associated with Matrigel or other ECM-derived hydrogels, there is a need for the development of chemically defined materials for cell culture and 3D printing^{39,40}. Our droplet-based technique can also print other hydrogels, such as collagen and agarose⁴¹. This flexibility could potentially offer the printing of 3D tissues containing different hydrogels in the future.

At P14L421, additional discussion was added: “The utilization of mouse sarcoma-derived Matrigel presents potential challenges in clinical applications, such as batch-to-batch variation, limited supply, and safety concerns arising from its tumor-derived origin. Hydrogels sourced from the ECM of specific tissues have been reported as viable alternatives to Matrigel for organoid production³⁸. However, the composition and structure of these hydrogels may still exhibit variation from one batch to another. To overcome the limitations associated with Matrigel or other ECM-derived hydrogels, chemically defined hydrogels for cells and organoids have been reported^{39,40}. Our droplet-based method is capable of printing alternative hydrogels, such as collagen and agarose⁴¹. This versatility holds the potential for future printing of 3D tissues comprising diverse and chemically defined hydrogels.”

Reference

- 1 Alcinesio, A. *et al.* Modular Synthetic Tissues from 3D-Printed Building Blocks. *Advanced Functional Materials* **32**, 2107773, doi:<https://doi.org/10.1002/adfm.202107773> (2022).
- 2 Zhou, L. *et al.* Lipid-Bilayer-Supported 3D Printing of Human Cerebral Cortex Cells Reveals Developmental Interactions. *Advanced Materials* **32**, 2002183, doi:<https://doi.org/10.1002/adma.202002183> (2020).
- 3 Alcinesio, A. *et al.* Controlled packing and single-droplet resolution of 3D-printed functional synthetic tissues. *Nature Communications* **11**, 2105, doi:10.1038/s41467-020-15953-y (2020).
- 4 Villar, G., Graham, A. D. & Bayley, H. A tissue-like printed material. *Science* **340**, 48-52, doi:10.1126/science.1229495 (2013).
- 5 Zhou, L. *et al.* Lipid-Bilayer-Supported 3D Printing of Human Cerebral Cortex Cells Reveals Developmental Interactions. *Adv Mater* **32**, e2002183, doi:10.1002/adma.202002183 (2020).
- 6 Ko, H. *et al.* Functional specificity of local synaptic connections in neocortical networks. *Nature* **473**, 87-91, doi:10.1038/nature09880 (2011).
- 7 Björklund, A. & Lindvall, O. Self-repair in the brain. *Nature* **405**, 893-895, doi:10.1038/35016175 (2000).
- 8 Haenseler, W. *et al.* A Highly Efficient Human Pluripotent Stem Cell Microglia Model Displays a Neuronal-Co-culture-Specific Expression Profile and Inflammatory Response. *Stem Cell Reports* **8**, 1727-1742, doi:10.1016/j.stemcr.2017.05.017 (2017).
- 9 Shi, Y., Kirwan, P. & Livesey, F. J. Directed differentiation of human pluripotent stem cells to cerebral cortex neurons and neural networks. *Nature Protocols* **7**, 1836-1846, doi:10.1038/nprot.2012.116 (2012).
- 10 Boissart, C. *et al.* Differentiation from human pluripotent stem cells of cortical neurons of the superficial layers amenable to psychiatric disease modeling and high-throughput drug screening. *Transl Psychiatry* **3**, e294, doi:10.1038/tp.2013.71 (2013).
- 11 Burke, E. E. *et al.* Dissecting transcriptomic signatures of neuronal differentiation and maturation using iPSCs. *Nature Communications* **11**, 462, doi:10.1038/s41467-019-14266-z (2020).
- 12 Galgano, M. *et al.* Traumatic Brain Injury: Current Treatment Strategies and Future Endeavors. *Cell Transplant* **26**, 1118-1130, doi:10.1177/0963689717714102 (2017).
- 13 Linaro, D. *et al.* Xenotransplanted Human Cortical Neurons Reveal Species-Specific Development and Functional Integration into Mouse Visual Circuits. *Neuron* **104**, 972-986.e976, doi:10.1016/j.neuron.2019.10.002 (2019).
- 14 Espuny-Camacho, I. *et al.* Human Pluripotent Stem-Cell-Derived Cortical Neurons Integrate Functionally into the Lesioned Adult Murine Visual Cortex in an Area-Specific Way. *Cell Rep* **23**, 2732-2743, doi:10.1016/j.celrep.2018.04.094 (2018).
- 15 Revah, O. *et al.* Maturation and circuit integration of transplanted human cortical organoids. *Nature* **610**, 319-326, doi:10.1038/s41586-022-05277-w (2022).

- 16 Morizane, A. *et al.* Direct Comparison of Autologous and Allogeneic Transplantation of iPSC-Derived Neural Cells in the Brain of a Nonhuman Primate. *Stem Cell Reports* **1**, 283-292, doi:<https://doi.org/10.1016/j.stemcr.2013.08.007> (2013).
- 17 Araki, R. *et al.* Negligible immunogenicity of terminally differentiated cells derived from induced pluripotent or embryonic stem cells. *Nature* **494**, 100-104, doi:10.1038/nature11807 (2013).
- 18 Sandor, C. *et al.* Transcriptomic profiling of purified patient-derived dopamine neurons identifies convergent perturbations and therapeutics for Parkinson's disease. *Human Molecular Genetics* **26**, 552-566, doi:10.1093/hmg/ddw412 (2017).
- 19 Frost, S., Barbay, S., Friel, K., Plautz, E. & Nudo, R. Reorganization of remote cortical regions after ischemic brain injury: a potential substrate for stroke recovery. *Journal of neurophysiology* **89**, 3205-3214 (2003).
- 20 Hatiboglu, M. A., Wildrick, D. M. & Sawaya, R. The role of surgical resection in patients with brain metastases. *Ecancermedicalscience* **7** (2013).
- 21 Cascino, G. D. Surgical treatment for epilepsy. *Epilepsy research* **60**, 179-186 (2004).
- 22 Schretlen, D. J. & Shapiro, A. M. A quantitative review of the effects of traumatic brain injury on cognitive functioning. *International review of psychiatry* **15**, 341-349 (2003).
- 23 Sun, J.-H., Tan, L. & Yu, J.-T. Post-stroke cognitive impairment: epidemiology, mechanisms and management. *Annals of translational medicine* **2** (2014).
- 24 Veretennikoff, K., Walker, D., Biggs, V. & Robinson, G. Changes in cognition and decision making capacity following brain tumour resection: illustrated with two cases. *Brain sciences* **7**, 122 (2017).
- 25 Elger, C. E., Helmstaedter, C. & Kurthen, M. Chronic epilepsy and cognition. *The Lancet Neurology* **3**, 663-672 (2004).
- 26 Kozlowski, D. A., Leasure, J. L. & Schallert, T. The control of movement following traumatic brain injury. *Comprehensive Physiology* **3**, 121-139 (2013).
- 27 Alarcón, F., Zijlmans, J., Duenas, G. & Cevallos, N. Post-stroke movement disorders: report of 56 patients. *Journal of Neurology, Neurosurgery & Psychiatry* **75**, 1568-1574 (2004).
- 28 Duffau, H. Acute functional reorganisation of the human motor cortex during resection of central lesions: a study using intraoperative brain mapping. *Journal of Neurology, Neurosurgery & Psychiatry* **70**, 506-513 (2001).
- 29 Tinuper, P. *et al.* Movement disorders in sleep: guidelines for differentiating epileptic from non-epileptic motor phenomena arising from sleep. *Sleep medicine reviews* **11**, 255-267 (2007).
- 30 McDonald, S., Togher, L. & Code, C. *Social and communication disorders following traumatic brain injury.* (Psychology press, 2013).
- 31 Brady, M. C., Kelly, H., Godwin, J., Enderby, P. & Campbell, P. Speech and language therapy for aphasia following stroke. *Cochrane database of systematic reviews* (2016).
- 32 Collée, E., Vincent, A., Dirven, C. & Satoer, D. Speech and language errors during awake brain surgery and postoperative language outcome in glioma patients: a systematic review. *Cancers* **14**, 5466 (2022).
- 33 Janszky, J. *et al.* Epileptic activity influences the speech organization in medial temporal lobe epilepsy. *Brain* **126**, 2043-2051 (2003).

- 34 Iaccarino, C., Carretta, A., Nicolosi, F. & Morselli, C. Epidemiology of severe traumatic brain injury. *J Neurosurg Sci* **62**, 535-541, doi:10.23736/S0390-5616.18.04532-0 (2018).
- 35 Dewan, M. C. *et al.* Estimating the global incidence of traumatic brain injury. *J Neurosurg*, 1-18, doi:10.3171/2017.10.JNS17352 (2018).
- 36 Song, S. Y., Lee, S. K., Eom, K. S. & Investigators, K. Analysis of Mortality and Epidemiology in 2617 Cases of Traumatic Brain Injury : Korean Neuro-Trauma Data Bank System 2010-2014. *J Korean Neurosurg Soc* **59**, 485-491, doi:10.3340/jkns.2016.59.5.485 (2016).
- 37 Hall, K. M. Establishing a national traumatic brain injury information system based upon a unified data set. *Archives of physical medicine and rehabilitation* **78**, S5-S11 (1997).
- 38 Kim, S. *et al.* Tissue extracellular matrix hydrogels as alternatives to Matrigel for culturing gastrointestinal organoids. *Nature Communications* **13**, 1692, doi:10.1038/s41467-022-29279-4 (2022).
- 39 Chrisnandy, A., Blondel, D., Rezakhani, S., Broguiere, N. & Lutolf, M. P. Synthetic dynamic hydrogels promote degradation-independent in vitro organogenesis. *Nature Materials* **21**, 479-487, doi:10.1038/s41563-021-01136-7 (2022).
- 40 Gjorevski, N. *et al.* Designer matrices for intestinal stem cell and organoid culture. *Nature* **539**, 560-564, doi:10.1038/nature20168 (2016).
- 41 Krishna Kumar, R. *et al.* Droplet printing reveals the importance of micron-scale structure for bacterial ecology. *Nature Communications* **12**, 857, doi:10.1038/s41467-021-20996-w (2021).

REVIEWER COMMENTS

Reviewer #1 (Remarks to the Author):

Most of my previous concerns were fully addressed and I am convinced that this article is ready for acceptance to Nature Communications if following minor concerns are to be resolved. They are

- Authors need to either circle or box OXF region with red line in Extended Data Fig1 that corresponds the Fig. 1L OXF area. It is more recommended to have a front image, not inverted one, however, that is optional.

- I would like to double-check that your group's previous works on DIBs were fully cited in the manuscript. If so, those changes should be reflected in main body of the manuscript, not as in the Response letter only. I was not able to find them in the manuscript.

- The schematic for your printing set-up has to be contained in supplementary data sheet. I found only the Table, but for the readers to fully grasp how your printing works, your protocols have to be visualized as an image.

Reviewer #2 (Remarks to the Author):

Overall, the authors have addressed several important concerns from the initial submission. I feel the manuscript is improved and more easily interpretable. In particular, the new revision to Figure 5 and the corresponding language to more clearly state which experiments involved implantation of deep layer or upper layer neurons helps with comprehension. Additionally, the authors investigated the host glial populations near the implantation site via immunostaining, which provides nice evidence of astrocyte infiltration and no overt inflammatory response.

However, there remain several concerns that I would like to mention again here.

1. In response to a question about the number of iPSC cell lines used for each experiment, the authors stated that they used three cell lines: “unlabelled AH016-3 hiPSCs, RFP-labelled AH016-3 hiPSCs and GFP-labelled AH016-3”. These three are all derived from the same patient sample, and are thus one single iPSC line. Ideally these experiments would have been performed with additional cell lines. Without these additional replicates, this manuscript lacks biological replicates. This is an important weakness. Although replicating these complicated experiments is a large undertaking, this fact should at the very least be addressed in the discussion.

2. In response to major concern 4 and minor concern 2, in which the authors were asked to validate their neuronal differentiations more deeply via RNA-sequencing or further antibody staining. The authors did not provide a compelling justification for not pursuing these experiments. The host species antibody limitation is a dubious reason to not further profile the neuronal populations. If the goal of this manuscript is to improve upon previous methods of implanting neurons to treat brain injury, the neuronal differentiation system should be well-validated and compared to the previously-published literature to ensure reproducibility and similarity to in vivo neurons. It's well acknowledged by this reviewer that these protocols are characterized and cited, but that does not ensure similar outcomes in these specific data.

Reviewer #3 (Remarks to the Author):

The authors provided additional context to the initial review. Yet, the revisions surrounding reframing the applications to not focus solely on TBI were modest. The authors revised one section of their introduction. The claims remain overstated since the experiments were conducted ex vivo. This criticism is regarding the framing and impact of the results. If the manuscript highlights the fundamental questions that were addressed in this manuscript rather than emphasizing and extrapolating the translational impact, then the manuscript would be appropriate.

Additionally, while edits were made regarding "functional" integration - the end characterization of electrophysiology, assessment of cell type after implantation, and secondary non-IHC measurements were still lacking from this manuscript. As such, while this manuscript is rigorous in terms of materials development, the biological assessments were less than requested in the first review.

REVIEWER COMMENTS

Reviewer #1 (Remarks to the Author):

Most of my previous concerns were fully addressed and I am convinced that this article is ready for acceptance to Nature Communications if following minor concerns are to be resolved. They are

- Authors need to either circle or box OXF region with red line in Extended Data Fig1 that corresponds the Fig. 1L OXF area. It is more recommended to have a front image, not inverted one, however, that is optional.

We have put a red box covering 'OXF' in the Extended Data Fig 1h to highlight that it is a magnification of the Fig. 1L 'OXF' area. In addition, we also adjusted the Extended Data Fig 1h image so that it is not inverted.

h

Revised Extended Data Fig 1h). A magnified view of the droplet network in Fig. 1l. The network was printed as 'OXF' and is indicated by the red box.

- I would like to double-check that your group's previous works on DIBs were fully cited in the manuscript. If so, those changes should be reflected in main body of the manuscript, not as in the Response letter only. I was not able to find them in the manuscript.

We thank the reviewer for pointing this out. We have now cited our group's previous work on DIBs in the manuscript.

At Lines P3L81 and L84, additional citations about DIBs have been added.

- The schematic for your printing set-up has to be contained in supplementary data sheet. I found only the Table, but for the readers to fully grasp how your printing works, your protocols have to be visualized as an image.

We agree that a graphic will help readers to better understand the printing process. Accordingly, we have composed a flowchart (Revised Supplementary Figure 1) to summarize the steps involved in our printing technique.

Revised Supplementary Figure 1. A flow chart of the droplet-based 3D bioprinting process¹⁻³.

At P4L95 "A flow chart of the printing process is shown in Supplementary Figure 1." Was added.

Reviewer #2 (Remarks to the Author):

Overall, the authors have addressed several important concerns from the initial submission. I feel the manuscript is improved and more easily interpretable. In particular, the new revision to Figure 5 and the corresponding language to more clearly state which experiments involved implantation of deep layer or upper layer neurons helps with comprehension. Additionally, the authors investigated the host glial populations near the implantation site via immunostaining, which provides nice evidence of astrocyte infiltration and no overt inflammatory response.

However, there remain several concerns that I would like to mention again here.

1. In response to a question about the number of iPSC cell lines used for each experiment, the authors stated that they used three cell lines: “unlabelled AH016-3 hiPSCs, RFP-labelled AH016-3 hiPSCs and GFP-labelled AH016-3”. These three are all derived from the same patient sample, and are thus one single iPSC line. Ideally these experiments would have been performed with additional cell lines. Without these additional replicates, this manuscript lacks biological replicates. This is an important weakness. Although replicating these complicated experiments is a large undertaking, this fact should at the very least be addressed in the discussion.

The reviewer is correct that the AH016-3 hiPSCs, despite the differences in fluorescent labelling, are derived from a single hiPSC line. Although only the AH016-3 hiPSC line was employed, the differentiation process has been repeated multiple times, demonstrating its replicability. We have revised both the Discussion and Methods sections to reflect that our experiments were performed with a single hiPSC cell line, AH016-3.

At Lines P14L425, " In the present study, our primary emphasis has been on the implantation of tissues prepared by our 3D printing technique, and only one hiPSC line, AH016-3, was tested. Nonetheless, the differentiation process was applied to these hiPSCs multiple times with similar results." was added.

At line P15 L444, “We used three AH016-3 hiPSC lines: unlabelled, RFP-labelled and GFP-labelled AH016-3 hiPSCs.” was changed to “We used a single AH016-3 hiPSC line without or with fluorescent labelling: unlabelled, RFP-labelled and GFP-labelled.”.

2. In response to major concern 4 and minor concern 2, in which the authors were asked to validate their neuronal differentiations more deeply via RNA-sequencing or further antibody staining. The authors did not provide a compelling justification for not pursuing these experiments. The host species antibody limitation is a dubious reason to not further profile the neuronal populations. If the goal of this manuscript is to improve upon previous methods of implanting neurons to treat brain injury, the neuronal differentiation system should be well-validated and compared to the previously-published literature to ensure reproducibility and

similarity to *in vivo* neurons. It's well acknowledged by this reviewer that these protocols are characterized and cited, but that does not ensure similar outcomes in these specific data.

In this study, we have reaffirmed the neuronal characterization by using valid biomarkers for immunofluorescence staining for CUX1, BRN2, SATB2, CTIP2, SOX2 and TUJ1, We have also used qPCR for CUX1, CUX2, BRN2, CTIP2, PAX6, and NESTIN. Our data are in agreement with those of Shi and Boissart ^{4,5}, who established the protocols for neural differentiation.

Our findings corroborate the generation of two distinct populations of hiPSC-derived neurons, one exhibiting deep-layer features and the other upper-layer features. We acknowledge that RNA sequencing could provide a more comprehensive profile of these neurons in comparison with their *in vivo* counterparts. However, the primary focus of our study is to demonstrate that a droplet-based 3D bioprinting technique, can generate two-layer living neural tissues that can be implanted into damaged brain tissue. In the future, neurons from more advanced neural differentiation protocols might be used to generate 3D printed tissues with more layers containing purer cells.

In clinical applications, it is anticipated that cells derived from patients will be reprogrammed for printing and implantation. Therefore, transcriptomic profiling of the current pre-implantation neurons, which are used here to demonstrate a procedure, would not provide further insights into future applications, where transcriptomics will indeed provide valuable information.

Reviewer #3 (Remarks to the Author):

The authors provided additional context to the initial review. Yet, the revisions surrounding reframing the applications to not focus solely on TBI were modest. The authors revised one section of their introduction. The claims remain overstated since the experiments were conducted *ex vivo*. This criticism is regarding the framing and impact of the results. If the manuscript highlights the fundamental questions that were addressed in this manuscript rather than emphasizing and extrapolating the translational impact, then the manuscript would be appropriate.

We have revised our manuscript to underscore our primary objective: the demonstration of a droplet-based 3D bioprinting technique, capable of fabricating biologically viable, two-layer cerebral cortical tissues. We have also carefully refined the scope of our writing on the translational implications of our work, streamlining the focus in line with the core purpose of this study.

The modifications of the manuscript are listed below.

1. Title was changed to "Integration of 3D-Printed Cerebral Cortical Tissue into an *ex vivo* Lesioned Brain Slice"

2. At P1L30, P3L63, P3L76, P9L260, P9L263, P11L321, P11L327 and P12L367, the term “ex vivo” was added.
3. The sentence in the abstract (P1L20) “Engineering human tissue with diverse cell types and desired cellular architectures and functions is a considerable challenge.” was changed to “Engineering human tissue with diverse cell types and desired cellular architectures **and functions** is a considerable challenge.
4. P1L34 in the Abstract was changed to “Importantly, **our methodology offers a technical reservoir for future personalized implantation treatments that use 3D tissues derived from a patient's own iPSCs.**”
5. At P2L44 in the Introduction, “Here, we focus on the generation of neural tissues for implantation, although our technology is widely applicable.” was changed to “Here, we focus on the generation of neural tissues **with laminar structures for potential applications involving implantation.**”
6. At P11L322, “Effective therapies using stem cell-derived cortical tissue for brain repair has not been established. A critical challenge is the difficulty in forming functional connections between implants and the host brain.” was deleted.
7. At P12L341, “To evaluate the functionality of the implanted cortical tissues, we performed Ca²⁺ imaging with Fluo-4” was changed to “To evaluate the **activity** of the implanted cortical tissues, we performed Ca²⁺ imaging with Fluo-4”.
8. At P13L379, “for better understanding the mystery of how intracortical human neuron circuits develop and lead to higher cognition” was deleted.
9. P14L410 was changed to “In the present study, **we demonstrated a droplet-based 3D bioprinting technique, capable of generating two-layer cortical tissues that can be implanted into mouse ex vivo brain explants. In the implanted explants,** we observed structural integration, based on implant-to-host process outgrowth, neuron migration, and correlated Ca²⁺ oscillations between the implant and the host.”

Additionally, while edits were made regarding "functional" integration - the end characterization of electrophysiology, assessment of cell type after implantation, and secondary non-IHC measurements were still lacking from this manuscript. As such, while this manuscript is rigorous in terms of materials development, the biological assessments were less than requested in the first review.

While, we understand reviewer’s concerns regarding the biological assessments, our study focuses on the development and application of a droplet-based 3D bioprinting technique. We

have further adjusted our manuscript to highlight technique development and the fundamental findings on printed tissues, without unwarranted extrapolation to potential clinical impact.

We believe the current level of biological validation is proportionate to our primary objective and supports our technical findings. Detailed biological evaluations, such as electrophysiology and secondary non-IHC measurements, will be more important for the characterisation of *in vivo* implants, in which there would be a more realistic environment for functional integration with the host. We are currently pursuing these long-term *in vivo* experiments.

References

- 1 Villar, G., Graham, A. D. & Bayley, H. A tissue-like printed material. *Science* **340**, 48-52, doi:10.1126/science.1229495 (2013).
- 2 Krishna Kumar, R. *et al.* Droplet printing reveals the importance of micron-scale structure for bacterial ecology. *Nature Communications* **12**, 857, doi:10.1038/s41467-021-20996-w (2021).
- 3 Zhou, L. *et al.* Lipid-Bilayer-Supported 3D Printing of Human Cerebral Cortex Cells Reveals Developmental Interactions. *Advanced Materials* **32**, 2002183, doi:<https://doi.org/10.1002/adma.202002183> (2020).
- 4 Shi, Y., Kirwan, P. & Livesey, F. J. Directed differentiation of human pluripotent stem cells to cerebral cortex neurons and neural networks. *Nature Protocols* **7**, 1836-1846, doi:10.1038/nprot.2012.116 (2012).
- 5 Boissart, C. *et al.* Differentiation from human pluripotent stem cells of cortical neurons of the superficial layers amenable to psychiatric disease modeling and high-throughput drug screening. *Transl Psychiatry* **3**, e294, doi:10.1038/tp.2013.71 (2013).

REVIEWERS' COMMENTS

Reviewer #2 (Remarks to the Author):

The authors' response to the most recent critiques from this referee are minimal. In general, it appears there is a reasonable disagreement on the level of biological characterization required for this study. The authors explain that they believe the point of the manuscript is simply to show two layer neural tissues can be implanted into brain tissue, and thus further characterization is not warranted. My personal opinion is that these details are relevant because it's important to understand exactly which cells are or are not properly integrating upon transplantation.

Ultimately, I feel this decision should be left to the editorial team since this is simply a fundamental difference of opinion.

Reviewer #3 (Remarks to the Author):

The authors addressed the comments/critiques adequately by revising the impact statement to focus on the key findings of their data and not extrapolate. The functional assessment still remains a hole in their experimental design and is appropriate for in vitro/ex vivo testing, contrary to the authors rebuttal. Yet, the materials development and platform technology is robust and impactful for the field.

REVIEWERS' COMMENTS

Reviewer #2 (Remarks to the Author):

The authors' response to the most recent critiques from this referee are minimal. In general, it appears there is a reasonable disagreement on the level of biological characterization required for this study. The authors explain that they believe the point of the manuscript is simply to show two layer neural tissues can be implanted into brain tissue, and thus further characterization is not warranted. My personal opinion is that these details are relevant because it's important to understand exactly which cells are or are not properly integrating upon transplantation.

Ultimately, I feel this decision should be left to the editorial team since this is simply a fundamental difference of opinion.

To address Reviewer's concerns, we have further toned down the functional claims within the text as stated above. Since the primary focus of our study is to demonstrate that our droplet-based 3D bioprinting technique can generate two-layer cortical tissues for implantation into damaged brain tissue and that cells derived from patients might be used in clinical applications,

intensive biological characterisation of the iPSC-derived neurons used in this study would not provide further insight.

Reviewer #3 (Remarks to the Author):

The authors addressed the comments/critiques adequately by revising the impact statement to focus on the key findings of their data and not extrapolate. The functional assessment still remains a hole in their experimental design and is appropriate for *in vitro*/*ex vivo* testing, contrary to the authors rebuttal. Yet, the materials development and platform technology is robust and impactful for the field.

We are glad the reviewer favours the technology developed in this work, which is our key focus. In our previous round of responses to the reviewers, we made modifications in multiple places to tone down our claims of functionality. We have now made two additional modifications, as stated above.

We believe our biological characterisation supports our technical findings and are appropriate to our goals. Further functional assessment, such as electrophysiology, will be important for the characterisation of *in vivo* implants, which we are currently pursuing.